# Meta Automatic Curriculum Learning for Classrooms of Deep RL Black-Box Students

## Abstract

A major challenge in the Deep RL (DRL) community is to train agents able to generalize their control policy over situations never seen in training. Training on diverse tasks has been identified as a key ingredient for good generalization, which pushed researchers towards using rich procedural task generation systems controlled through complex continuous parameter spaces. In such complex continuous task spaces, it is essential to rely on some form of Automatic Curriculum Learning (ACL) to adapt the task sampling distribution to a given learning agent, instead of randomly sampling tasks, as many could end up being either trivial or unfeasible. Since it is hard to get prior knowledge on such task spaces, many ACL algorithms explore the task space to detect progress niches over time. This costly tabula-rasa search process needs to be performed for each new learning agents, although they might have similarities in their capabilities profiles. To address this limitation, we introduce the concept of Meta-ACL, and formalize it in the context of black-box RL learners, i.e. algorithms seeking to generalize curriculum generation to an (unknown) distribution of learners. In this work, we present AGAIN, a first instantiation of Meta-ACL, and showcase its benefits for curriculum generation over classical ACL in multiple simulated environments including procedurally generated parkour environments with learners of varying morphologies. Videos and code are available at `https://sites.google.com/view/meta-acl`.

## 1 Introduction

The idea of organizing the learning sequence of a machine is an old concept that stems from multiple works in reinforcement learning (Selfridge et al., 1985; Schmidhuber, 1991), developmental robotics (Weng et al., 2001; Oudeyer et al., 2007) and supervised learning (Elman, 1993; Bengio et al., 2009), from which the Deep RL community borrowed the term *Curriculum Learning*. Automatic CL (Portelas et al., 2020) refers to *teacher* algorithms able to autonomously adapt their task sampling distribution to their evolving *student*. In DRL, ACL has been leveraged to scaffold learners in a variety of multi-task control problems, including video-games (Pathak et al., 2017; Burda et al., 2019; Salimans & Chen, 2018), multi-goal robotic arm manipulation (Andrychowicz et al., 2017; Colas et al., 2019; Cideron et al., 2019; Fournier et al., 2018) and navigation in sets of environments (Matiisen et al., 2017; Portelas et al., 2019; Mehta et al., 2019; Florensa et al., 2018; Klink et al., 2020). Concurrently, multiple authors demonstrated the benefits of Procedural Content Generation (PCG) as a tool to create rich task spaces to train generalist agents (Justesen et al., 2018; Risi & Togelius, 2019; Cobbe et al., 2019). However, a current limit of ACL is that, when applied to such large continuous task spaces, which often have few learnable subspaces, it either relies on human expert knowledge that is hard/costly to provide (and which undermines how automatic the ACL approach is), or it loses a lot of time finding tasks of appropriate difficulty through *task exploration*. This random search for progress niches over the task space is a costly *tabula rasa* process. While it seems acceptable to perform given a single agent to train, it becomes suboptimal whenever considering the training of multiple agents that might have similarities in their capabilities profiles.

Beyond training single DRL learners with ACL to generalize over task spaces, we propose to go further and work on training (unknown) distributions of students on continuous task spaces. We propose to use the term *Classroom Teaching* (CT) to refer to such problems. CT defines a family of problems in which a teacher

algorithm is tasked to sequentially generate multiple curricula tailored for each of its students, all having potentially varying abilities. CT differs from the problems studied in population-based developmental robotics (Forestier et al., 2017) and evolutionary algorithms Wang et al. (2019; 2020) as in CT there is no direct control over the characteristics of learners, and the objective is to foster maximal Learning Progress (LP) over all learners rather than iteratively populating a pool of high-performing task-expert policies. Studying CT scenarios brings DRL closer to assisted education research problems and might stimulate the design of methods that alleviate the expensive use of expert knowledge in this field (Koedinger et al., 2013; Clément et al., 2015).

Given multiple students to train, no expert knowledge, and assuming at least partial similarities between each students' optimal curriculum, current *tabula rasa* exploratory-ACL approaches that do not reuse knowledge between different students do not seem like the optimal choice. This motivates the research of what we propose to call *meta automatic curriculum learning* mechanisms, i.e. algorithms learning to generalize ACL over multiple students. In this work we formalize this novel setup and propose AGAIN, a first Meta-ACL baseline algorithm inspired from an existing ACL method (Portelas et al., 2019). Given a new student to train, our approach is centered on the extraction of adapted curriculum priors from a history of previously trained students. The prior selection is performed by matching competence vectors that are built for each student through pre-testing. We show that this simple method can bring significant performance improvements over classical ACL in both a toy environment without DRL students and in Box2D locomotion environments with DRL learners.

**Motivations** We argue that finding ways to efficiently train multiple agents is a valid research endeavor, which could find potential application areas. For example, in robotics, when considering a specific set of problems to solve, researchers often rely on iteratively modifying the morphology of their robot (e.g. they change motors, add degrees of freedom, change end-effector) based on the performance obtained by a given morphology on the considered task set. In such cases, the resulting sequence of agents to train bear strong similarities with each others. If the complexity of the task set requires to use curriculum generation procedures for each agent, the ability to leverage previous teaching data to bootstrap training seems more appropriate than brute-force task space exploration. Similar issues arise when considering the iterative enhancement of DRL architectures (regardless of embodiment). On a more speculative note, innovations in industrial automation might significantly rely on robotic systems that are iteratively trained (e.g. using DRL in simulation) to master a significant range of tasks. The sensorimotor apparatus of such systems might have to be modified to the specific environmental condition of each industrial customer. In such cases, providing sample efficient ways to train these set of robotic systems appears as an important component for economic viability.

**Main contributions:**

- Introduction to the concept of Meta-ACL, i.e. algorithms that generalize curriculum generation in Classroom Teaching scenarios, along with an approach to study these algorithms. Formalization of the interaction flows between Meta-ACL algorithms and (unknown) DRL student distributions.

- Introduction of AGAIN, a first Meta-ACL baseline algorithm which learns curriculum priors to fasten the identification of learning progress niches for new DRL students. Analysis of AGAIN on both a toy-environment and a parametric Box2D locomotion environment, demonstrating the performance advantages of this approach over classical ACL, including (and surprisingly) when applied to a single student.

## 2 Related Work

To approach the problem of curriculum generation for DRL agents, recent works proposed multiple ACL algorithms based on the optimization of surrogate objectives such as learning progress (Matiisen et al., 2017; Mysore et al., 2018; Colas et al., 2019; Portelas et al., 2019), diversity (Bellemare et al., 2016; Eysenbach et al., 2019; Jabri et al., 2019) or intermediate difficulty (Florensa et al., 2017; 2018; OpenAI et al., 2019;

Mehta et al., 2019; Racaniere et al., 2020). While such works focus on training students through independent runs, we propose to investigate how one can share information across multiple trainings. Within DRL, *policy distillation* (Teh et al., 2017; Czarnecki et al., 2019) consists in leveraging one or several previously trained policies to perform *behavior cloning* on a new policy (e.g. to speed up training and/or to leverage task-experts to train a multi-task policy). Our work can be seen as proposing a complementary toolbox, aiming to perform *curriculum distillation* on a continuous space of tasks.

Similar ideas were developed for supervised learning (Yim et al., 2017; Furlanello et al., 2018; Hacohen & Weinshall, 2019). Hacohen & Weinshall (2019) propose an approach to infer a curriculum from past training for an image classification task: they train their network once without curriculum and use its predictive confidence for each image as a difficulty measure exploited to derive an appropriate curriculum to re-train the network. Although we are mainly interested in training a classroom of diverse students, section 5.3 presents similar experiments in a DRL scenario, showing that our Meta-ACL procedure can be beneficial for a single learner that we train once and re-train using curriculum priors inferred from the first run.

Our work bears some similarities with Turchetta et al. (2020), which used a data-driven approach to autonomously infer curricula for multiple DRL agents. However, in their work, each agent is trained to perform a single navigation task: the "curriculum" consists in the selection of safety constraints to apply during training, chosen among a pre-defined discrete set (e.g. reset agent to previous state if it reaches a dangerous location). Agents only differ by their network initializations. By contrast, we consider the problem of training *generalist* students with varying morphologies (and network initializations), with a teacher algorithm choosing tasks from a *continuous* task space.

In Narvekar & Stone (2020), authors also propose to study how to generalize curriculum generation for policy learners. Given a space of navigation goals as independent target objectives in a grid-world environment, they show that they are able to efficiently generate training curricula for each new target goal presented to a SARSA (Rummery & Niranjan, 1994) policy learner. Curriculum generation is performed by a high-level DQN agent (Mnih et al., 2015), conditioned on the student's weights (i.e. its knowledge state) and the target goal, whose actions are to select on which source goal (9 predefined possibilities) to train the student. Compared to this work, we study how to generalize curriculum generation to *diverse* learners given a single space of task to master. We consider complex DRL students and locomotion environments with continuous actions.

## 3    Meta Automatic Curriculum Learning Framework

The following paragraphs formalize the concept of Meta-ACL for black-box students. More precisely, we extend existing frameworks for ACL (Portelas et al., 2020) and teacher-student interactions on continuous task spaces (Portelas et al., 2019) to introduce our Classroom Teaching setup.

**Continuous Task Space**    We use the term continuous task space to refer to a parameter space $\mathcal{T} \subset \mathbb{R}^n$ encoding the transition and reward functions along with the initial state distributions of Partially Observable Markov Decision Processes: any $\tau \in \mathcal{T}$ encodes the procedural generation of a distribution of environments, a.k.a. a distribution of tasks. For simplicity and compactness, we propose to refer to a distribution $\tau$ as a single *stochastic* task. As in (Portelas et al., 2019), minimal assumptions are made over $\mathcal{T}$: the difficulty profile of $\mathcal{T}$ is unknown and therefore assumed to be a piece-wise smooth function, potentially containing large unfeasible or trivial subspaces (w.r.t. to considered learners).

**Black-box students**    The Meta-ACL framework assumes the existence of policy learners, a.k.a students $s$, capable of interacting in episodic control tasks (which can be defined as POMDPs). These students are assumed non-resetable, as in classical ACL scenarios: it is not possible to re-initialize or restore a student's knowledge state in the middle of training (e.g. use backed-up neural network parameters in the advent of catastrophic forgetting). From an ACL perspective, this means the task sampling policy must be optimized in a single rollout. The student's objective is to maximize reward collection. Such students are confronted with tasks drawn from a continuous task space $\mathcal{T}$. We do not assume expert knowledge over this task space w.r.t students, e.g. task subspaces could be trivial for some students and unfeasible for others. The objective

of Meta-ACL is precisely to autonomously infer such prior knowledge from experience in scenarios where human expert knowledge is either hard or impossible to use (e.g. environments featuring complex PCG). Similarly, we consider *black-box* students, i.e. the internal states of students are not accessible to teachers, and we do not assume which learning mechanisms are used (e.g. SAC (Haarnoja et al., 2018), PPO (Schulman et al., 2017), evolutionary algorithms).

**Automatic Curriculum Learning**    Given a black-box student $s$ to train on a continuous task space $\mathcal{T}$, an ACL algorithm can be formalized as an algorithm producing a distribution $\mathcal{D}(\tau \mid \mathcal{H}_s^{int})$ which sequentially samples (parameterized) tasks for $s$ given its (growing) training history $\mathcal{H}_s^{int}$, i.e. the list of episodic rewards (a.k.a. returns) obtained for each task $s$ was trained on. Under an interaction budget of $E$ episodes ($E$ tasks) during which a student is allowed to interact with the environment, the performance of an ACL algorithm can be defined as the overall post-training competence of its student over $\mathcal{T}$:

$$\mathcal{C} = \int_{\mathcal{T}} c_\tau^E \, \mathrm{d}\tau, \tag{1}$$

with $c_\tau^E$ the post-training competence of the student on task $\tau$, which we simply define as the episodic reward. Since direct optimization of such a post-training performance is difficult, ACL is often approached using proxy objectives (e.g. learning progress, intermediate difficulty).

**Meta-ACL for Classroom Teaching.**    We now present the concept of Meta-ACL applied to a Classroom Teaching scenario, i.e. there is no longer a single student $s$ to be trained, but a set of students with varying abilities (e.g. due to morphology and/or learning mechanisms), sequentially drawn from an unknown student distribution $\mathcal{S}$. The concept of meta-learning refers to *any type of learning guided by prior experience with other tasks* (Vanschoren, 2018). In classical meta-RL settings, these "tasks" are defined as distinct MDPs. The novelty of our setup is to consider meta-learning at the level of teacher algorithms, for which *tasks* correspond to distinct *students* to train. In other words, Meta-ACL approaches must leverage knowledge from curricula built for previous students to improve the curriculum generation for new ones. More precisely, a Meta-ACL algorithm can be formulated as producing the following distribution:

$$\hat{\mathcal{D}}(\tau \mid \mathcal{H}_{s^K}^{int}, X) \quad s.t. \quad X = f(\mathcal{H}^{\mathcal{S}})$$
$$\mathcal{H}^{\mathcal{S}} = [\mathcal{H}_{s^0}^{int}, \mathcal{H}_{s^1}^{int}, ..., \mathcal{H}_{s^{K-1}}^{int}] \tag{2}$$

$s^K$ is the student being trained by $\hat{\mathcal{D}}$. $f$ is an algorithm extracting curriculum priors $X$ over a history $\mathcal{H}^{\mathcal{S}}$ of past K student trainings, resulting from the teaching of $K$ previous students with an ACL or Meta-ACL policy. We denote by *curriculum priors* any form of information relevant to augment a curriculum generation mechanism, e.g. a set of important tasks to focus on (i.e. a list of vectors) or – as in our experiments – a list of Gaussians. Given our formalization of ACL (see eq. 1), the experimenter's performance measure for Meta-ACL can be expressed as follows:

$$\hat{\mathcal{C}} = \int_{\mathcal{S}} \int_{\mathcal{T}} c_{s,\tau}^E \, \mathrm{d}\tau \, \mathrm{d}s. \tag{3}$$

As in the case of the ACL, direct optimization to maximize this performance measure is difficult as it implies the joint maximization of multiple students' performance. In our experiments, we reduce the Meta-ACL problem to the sequential independent training of a finite set of new students by leveraging priors from previous student trainings (with the hope to maximize performance over the entire set). Figure 1 provides a visual transcription of the workflow of a Meta-ACL algorithm. While in our experiments we use a fixed-size history $\mathcal{H}^{\mathcal{S}}$ of ACL-trained "previous" students (to make our experiments computationally tractable), $\mathcal{H}^{\mathcal{S}}$ could be grown incrementally by collecting training trajectories online from Meta-ACL trainings.

## 4    AGAIN: a First Meta-ACL Baseline

In this section, we present AGAIN (Alp-Gmm And Inferred Progress Niches), our proposed Meta-ACL algorithm, and connect it to the formalism described in section 3. In essence, AGAIN is based on characterizing

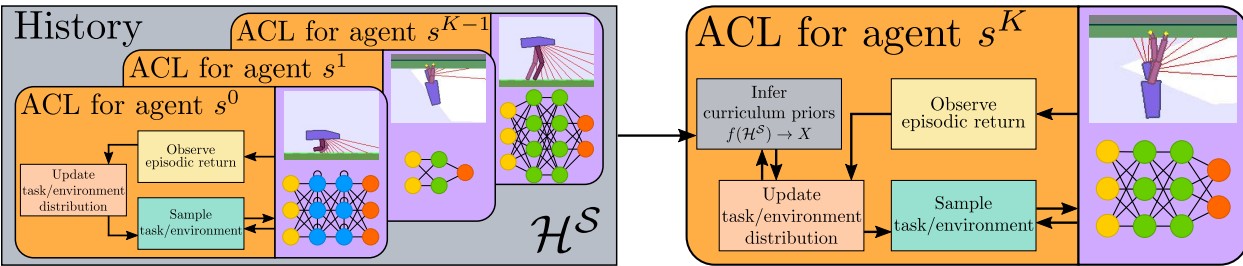

Figure 1: In *Meta Automatic Curriculum Learning* (Meta-ACL), the objective is to leverage previous teaching experience to improve the curriculum generation of a new agent, whose embodiment and learning mechanisms have potentially never been seen before: the teacher has to *generalize* over students.

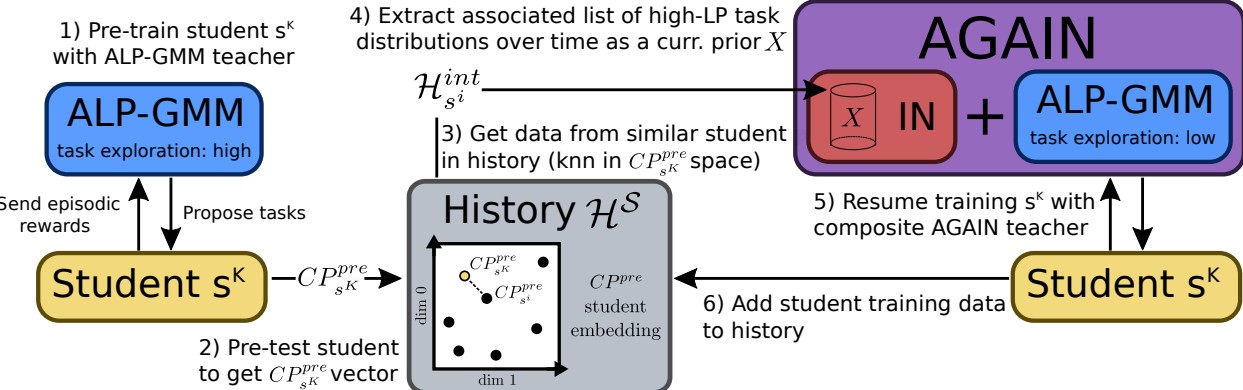

Figure 2: Schematic pipeline of AGAIN, our proposed Meta-ACL approach. Given a new student, AGAIN first performs a preliminary run with a high-exploration ALP-GMM curriculum generator. It then pre-test its new student and compare it to previous ones to infer an expert curriculum (IN). The training of the new student is then resumed with a combination of IN and a low-exploration ALP-GMM.

the performance of a new student, finding the most similar (performance-wise) previously trained student, and extracting promising curriculum data to resume teaching the new student. We first give a broad overview of the approach and then provide detailed explanations of key components.

**Overview**   Figure 2 provides a schematic pipeline of our Meta-ACL approach. Given a history $\mathcal{H}^{\mathcal{S}}$ of previously trained students and a new student $s^K \sim \mathcal{S}$ to train, AGAIN starts by (1) pre-training $s^K$ using ALP-GMM, an existing ACL algorithm from Portelas et al. (2019), well suited for teaching scenarios without expert knowledge. After this pre-training, AGAIN (2) challenges the student with a set of test tasks to construct a multidimensional *Competence Profile* $CP^{pre}$ of the student. This vector is then (3) used to select a previously trained student $s^i$ similar to $s^K$ from which the training history $\mathcal{H}^{int}_{s^i}$ is recovered. Based on $\mathcal{H}^{int}_{s^i}$, (4) AGAIN infers a set of curriculum priors $X$ (a list of high-LP task subspaces). Finally, (5) the training of $s^K$ can resume using a composite curriculum generator using both an expert curriculum derived from $X$ (for exploitation), and ALP-GMM (for exploration). The following sections provide in-depth details for each aforementioned step of our method.

## 4.1   ALP-GMM (1)

ALP-GMM is a LP-based ACL technique for continuous task spaces that does not assume prior knowledge. In short, ALP-GMM frames the task sampling problem into a non-stationary Multi-Armed bandit setup (Auer et al., 2002) in which arms are Gaussians spanning over the task space (see figure 3). The utility of each Gaussian is defined with a local LP measure derived from episodic reward comparisons, i.e. from the training history $\mathcal{H}^{int}$. The essence of ALP-GMM is to periodically fit a Gaussian Mixture Model (GMM) on recently sampled task parameters *concatenated with their respective estimated LP*. The Gaussian from which to sample

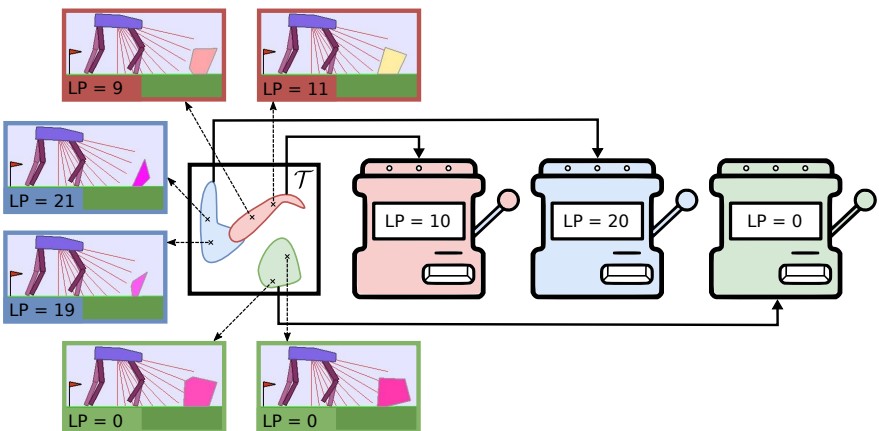

Figure 3: The core mechanism of ALP-GMM (Portelas et al., 2019) is to dynamically detect subspaces having different LP value within the task space. Each of these subspaces are used as an arm of a Multi Armed Bandit setup: their utility is computed using a local aggregated LP value. More precisely, they consider *absolute* LP, as learning regress equates to forgetting, which is valuable information indicating that re-training on the subspace must be done.

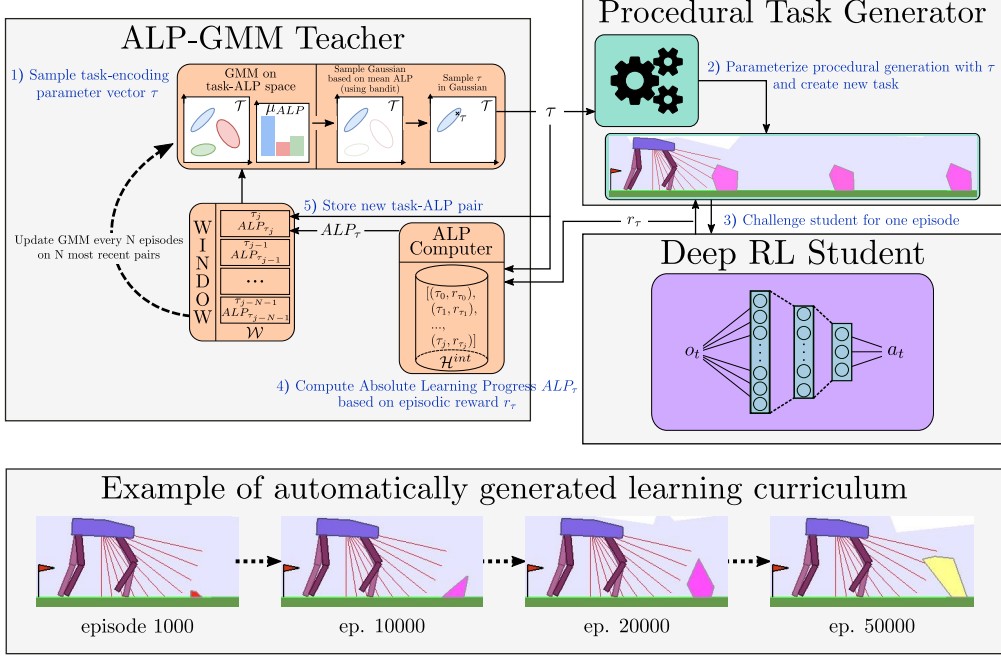

Figure 4: Schematic view of an ALP-GMM teacher's workflow from Portelas et al. (2019)

a new task is chosen proportionally to its mean LP dimension. Task exploration happens initially through a bootstrapping period of random task sampling and during training through residual (i.e. occasional) random task sampling. See figure 4 for a schematic pipeline of ALP-GMM and algorithm 1 for pseudo-code.

**Absolute LP computation** ALP-GMM relies on an empirical per-task computation of Absolute Learning Progress (ALP), allowing to fit a GMM on a concatenated space composed of tasks' parameters and respective ALP. Given a new task-encoding vector $\tau_{new} \in \mathcal{T} \subset \mathbb{R}^n$ on which the student's policy collected the episodic reward $r_{new} \in \mathbb{R}$, its ALP is computed using the closest previous task $\tau_{old}$ (Euclidean distance) with associated episodic reward $r_{old}$:

$$ALP_{\tau_{new}} = |r_{\tau_{new}} - r_{\tau old}| \tag{4}$$

---

**Algorithm 1** Absolute Learning Progress Gaussian Mixture Model (ALP-GMM)

---

**Require:** Student $s$, bounded task space $\mathcal{T}$, probability of random sampling $p_{rnd}$, fitting rate $N$, max number of Gaussians $k_{max}$

1: Initialize task-ALP First-in-First-Out window $\mathcal{W}$, set max size to $N$
2: Initialize task-reward history database $\mathcal{H}^{int}$
3: **loop** $N$ times                                             ▷ Bootstrap phase
4:     Sample random task-encoding parameters $\tau \in \mathcal{T}$
5:     Generate environment with $\tau$, send it to $s$, observe episodic reward $r_\tau$
6:     Compute ALP of $\tau$ based on $r_\tau$ and $\mathcal{H}^{int}$ (see equation 4)
7:     Store $(\tau, r_\tau)$ pair in $\mathcal{H}^{int}$, store $(\tau, ALP_\tau)$ pair in $\mathcal{W}$
8: **loop**                                    ▷ Stop after sampling $E$ tasks (including bootstrap)
9:     Fit a set of GMM having 2 to $k_{max}$ kernels on $\mathcal{W}$
10:     Select the GMM with best Akaike Information Criterion
11:     **loop** $N$ times
12:         $p_{rnd}$% of the time, sample a random task $\tau \in \mathcal{T}$
13:         Else, sample $\tau$ from a Gaussian chosen proportionally to its mean ALP value
14:         Generate environment with $\tau$, send it to $s$, observe episodic reward $r_\tau$
15:         Compute ALP of $\tau$ based on $r_\tau$ and $\mathcal{H}^{int}$
16:         Store $(\tau, r_\tau)$ pair in $\mathcal{H}^{int}$, store $(\tau, ALP_\tau)$ pair in $\mathcal{W}$
17: **Return** $s$

---

All previously encountered task's parameters and their associated ALP, a.k.a. task-ALP for short, recorded in a history database $\mathcal{H}^{int}$, are used for this computation.

**GMM fitting and sampling**   Given a database of task-ALP pairs, ALP-GMM's main mechanism is to fit a GMM on the concatenated space of tasks and ALP, i.e. a dataset of vectors of $n+1$ dimensions. Using such a fitting process relying on task-specific $ALP_\tau$ estimates one can then obtain a set of $k$ Gaussians $\left(\mathcal{N}(\boldsymbol{\mu_i}, \boldsymbol{\Sigma_i})\right)_{i=1}^{k}$, with $\boldsymbol{\mu_i} \in \mathbb{R}^{n+1}$. Given this formulation, one can interpret the mean ALP dimension of $\boldsymbol{\mu_i}$ as a local aggregated ALP measure, which we propose to refer to as $ALP_i$. The fitting of the GMM is performed every $N$ episodes on a window $\mathcal{W}$ containing the $N$ most recent task-ALP pairs. To sample new tasks, ALP-GMM first samples a Gaussian of the GMM either at random with probability $\rho_{rnd}$ or proportionally to their local $ALP_i$ value, with the probability of each Gaussian $i$ expressed as $\frac{ALP_i}{\sum_j ALP_j}$.

To adapt the number of components of the GMM online, a batch of GMMs having from 2 to $k_{max}$ components is fitted on $\mathcal{W}$, and the best one, according to Akaike's Information Criterion Bozdogan (1987) (i.e. a measure of how accurate the GMM-model is w.r.t. the task-ALP data), is kept as the new GMM. In all of our experiments we use the same hyperparameters as in Portelas et al. (2019) ($N = 250$, $k_{max} = 10$), except for the percentage of random task sampling $\rho_{rnd}$ which we set to 10% (we found it to perform better than 20%) when running ALP-GMM. See algorithm 1 for pseudo-code and figure 4 for a schematic pipeline (both are adapted from Portelas et al. (2019)). Note that in the remainer of this paper we refer to ALP as LP for simplicity, i.e. $LP_{ti}$ in $\mathcal{X}_{raw}$ from eq. 5 is equivalent to the mean ALP of Gaussians ($ALP_i$) in ALP-GMM.

## 4.2 Curriculum priors: selection (2,3)

How to extract curriculum priors $X$ from previous teaching data $\mathcal{H}^{\mathcal{S}}$ ? In other words, how to implement algorithm $f$ from eq. 2? As a first step towards more end-to-end approaches, we rely on a multi-step procedure inspired from knowledge assessment in Intelligent Tutoring Systems setups studied in the educational data mining literature (Vie, 2016; Vie et al., 2018). Because our procedure relies on extracting a set of high learning-progress subspaces, it assumes that all students in $\mathcal{H}^{\mathcal{S}}$ were either trained with AGAIN or ALP-GMM.

For a new student $s^K$, given its capabilities on the considered task space, how to select the most relevant previously trained student in $\mathcal{H}^{\mathcal{S}}$, from which to extract curriculum priors ? We propose to use *pre-tests* to

derive a competence profile vector $CP^{pre} \in \mathbb{R}^m$ for all trained students. Each dimensions of $CP^{pre}$ contains the episodic return of the student on the corresponding pre-test task. Given that we do not assume access to expert knowledge, we build this pre-test task set by selecting $m$ tasks uniformly over the task space (and leave the automatic construction of adaptive task sets for future work). We use the same task set to build a post-training CP vector $CP^{post} \in \mathbb{R}^m$ whose dimensions are summed up to get a score $j_s \in \mathbb{R}$, used to evaluate the end performance of students in $\mathcal{H}^{\mathcal{S}}$. After the initial pre-training of $s^K$ with ALP-GMM, curriculum priors can be obtained in 3 steps:

1. pre-test $s^K$ to get its CP vector $CP^{pre}_{s^K}$ (i.e. compute its initial performance assessment),

2. infer the $k$ most similar previously trained students in $CP^{pre}$ space using a k-nearest neighbor algorithm (i.e. find previous students with similar initial performance assessment), and

3. use the training history $\mathcal{H}^{int}$ of the student with maximal post-training score $j_s$ among those $k$. By selecting the previous student with maximal post-training score, this step ensures that curriculum data will be collected from the student that experienced the most competence progress out of these $k$, i.e. a previous student that most likely benefited from an efficient curriculum.

In essence, this method is about re-using curriculum data from a similarly-skilled and successfully-trained student.

### 4.3 Curriculum priors: curation (4)

Let $s^i$ be the student from $\mathcal{H}^{\mathcal{S}}$ resulting from the aforementioned selection procedure for the new student $s^K$. Assuming ALP-GMM (or AGAIN) as the underlying teacher used for $s^i$, we can extract curriculum priors $X$ by considering the ordered sequence of GMMs $X_{raw}$ that were periodically fitted throughout training (i.e. throughout $\mathcal{H}^{int}_{s^i}$):

$$X_{raw} = \{p(1), ..., p(T)\}$$
$$s.t. \quad p(t) = \sum_{i=1} LP_{ti} \mathcal{N}(\boldsymbol{\mu_{ti}}, \boldsymbol{\Sigma_{ti}}), \tag{5}$$

with $T$ the total number of GMMs in the list and $LP_{ti}$ the Learning Progress of the $i^{th}$ Gaussian from the $t^{th}$ GMM. Since the LP value of each Gaussian can be considered as its utility, we propose a simple method to leverage $X_{raw}$: $X$ can be obtained from $X_{raw}$ by keeping only Gaussians with $LP_{ti}$ above a predefined threshold $\delta_{LP}$, which creates a curated list $X$ containing only Gaussians located on task subspaces on which $s^i$ experienced learning progress.

### 4.4 AGAIN (5)

Given that a curriculum prior $X$, in the form of a list of high-LP Gaussian over time has been constructed, how to use it to generate an improved curriculum for student $s^K$? We propose to leverage $X$ by deriving an "expert" curriculum from it, thereafter named Inferred progress Niches (IN).

**IN** Given a GMM of $X$, a task can be selected by 1) sampling a Gaussian proportionally to its $LP_{ti}$ value, and 2) sampling the Gaussian to obtain the task parameters. But how to decide which GMMs to use along the training of the new student $s^K$ ?

While the simplest way to obtain such a curriculum would be to start sampling tasks from the first GMM and step to the next GMM at the same rate as in the initial ALP-GMM run, we propose a more flexible *reward-based* method. This method requires us to record the list $\mathcal{R}_{raw}$ of mean episodic rewards obtained by the previously trained student $s^i$ for each GMM of $X$ (which can be done without additional assumptions or computational overhead):

$$\mathcal{R}_{raw} = \{\mu^1_r, ..., \mu^t_r, \mu^T_r\} \quad s.t \quad |\mathcal{R}_{raw}| = |X_{raw}|, \tag{6}$$

with T the total number of GMMs in the first run (same as in $X_{raw}$), and $\mu_r^t$ the mean episodic reward obtained by the first DRL agent during the last 50 tasks sampled from the $t^{th}$ GMM. Given this, to select which GMM from $X$ is used to sample tasks over time along the training of $s^K$, we start with the first GMM and only iterate over $X$ once the mean episodic reward over tasks recently sampled from the current GMM matches or surpasses the mean episodic reward recorded during the initial ALP-GMM run. In app. C, we show experimentally that this *reward-based* variant outperforms other potential methods.

**AGAIN**  Simply using IN directly for $s^K$ lacks adaptive mechanisms towards the characteristics of the new student (e.g. new embodiment, different initial parameters), which could lead to failure cases where the expert curriculum misses important aspects of training (e.g. detecting task subspaces that are being forgotten). Additionally, the meta-learned ACL algorithm must have the capacity to emancipate from the expert curriculum once the trajectory is completed (i.e. go beyond $X$). This motivates why our approach combines IN with an ALP-GMM teacher after the initial pre-training. A simple way to combine the changing GMM of IN and ALP-GMM over time is to build a GMM $G$ containing Gaussians from the current GMM of IN and ALP-GMM. By selecting the Gaussian in $G$ from which to sample a new task using their respective $LP$, this approach allows to adaptively modulate the task sampling between both, shifting the sampling towards IN when ALP-GMM does not detect high-LP subspaces and towards ALP-GMM when the current GMM of IN have lower-LP Gaussians. When the last GMM $p(T)$ of the IN curriculum is reached, we switch the fixed $LP_{Ti}$ values of all IN Gaussians to periodically updated LP estimates, i.e. we allow AGAIN to modulate the importance of $p(T)$ for task sampling depending on its current student's performance. See algorithms 2 3 and 4 for pseudo-code and appendix A for additional details on variant approaches and hyperparameter choices.

---

**Algorithm 2**  Pretrain phase (AGAIN helper algorithm)

---

**Require:** Student policy $s_\theta$, teacher training history $\mathcal{H}^\mathcal{S}$, task-encoding parameter space $\mathcal{T}$, LP threshold $\delta_{LP}$, experimental pre-train budget $E_{pre}$, pre-test set size $m$, number of neighbors for student selection $k$, random sampling ratio $\rho_{high}$

1: Init $s_\theta$, train it for $E_{pre}$ env. steps with ALP-GMM $(\rho_{high}, \mathcal{T})$                                        ▷ See algo. 1
2: Pre-test $s_\theta$ with $m$ tasks selected uniformly over $\mathcal{T}$ and get $CP_s^{pre}$                                       ▷ Pre-test phase
3: Apply knn algorithm in CP space of $\mathcal{H}^\mathcal{S}$, get $k$ students closest to $CP_s^{pre}$
4: Among those $k$, keep the one with highest summed post training $CP^{post}$, extract its $X_{raw}$
5: Get $X$ from $X_{raw}$ by removing any Gaussian with $LP_{ti} < \delta_{LP}$.
6: **Return** $X$

---

**Algorithm 3**  Inferred progress Niches (IN)

---

**Require:** Student policy $s_\theta$, teacher training history $\mathcal{H}^\mathcal{S}$, task-encoding parameter space $\mathcal{T}$, LP threshold $\delta_{LP}$, update rate $N$, experimental budget $E$, experimental pre-train budget $E_{pre}$, pre-test set size $m$, number of neighbors for student selection $k$, random sampling ratio $\rho_{high}$

1: Launch Pretrain phase and get expert GMM list $X$                                                        ▷ See algo. 2
2: Initialize reward First-in-First-Out window $\mathcal{W}$, set max size to $N$
3: Initialize expert curriculum index $i_c$ to 0
4: **loop** Stop after sampling $E$ tasks (including pre-train)
5:     If $\mathcal{W}$ is full, compute mean reward $\mu_w$ from $\mathcal{W}$
6:         If $\mu_w$ superior to $i_c^{th}$ reward threshold in $\mathcal{R}$, set $i_c$ to $min(i_c + 1, len(X))$
7:     Set current GMM $G_{IN}$ to $i_c^{th}$ GMM in $X$
8:     Sample $\tau$ from a Gaussian in $G_{IN}$ chosen proportionally to its $LP_{ti}$
9:     Generate env. with $\tau$, send it to student $s_\theta$ and add episodic reward $r_\tau$ to $\mathcal{W}$
10: Add student's training data to $\mathcal{H}^\mathcal{S}$
11: **Return** $s_\theta$

---

---

**Algorithm 4** Alp-Gmm And Inferred progress Niches (AGAIN)

---

**Require:** Student policy $s_\theta$, teacher training history $\mathcal{H}^S$, task-encoding parameter space $\mathcal{T}$, LP threshold $\delta_{LP}$, update rate $N$, experimental budget $E$, experimental pre-train budget $E_{pre}$, pre-test set size $m$, number of neighbors for student selection $k$, random sampling ratio $\rho_{low}$ and $\rho_{high}$

1: Launch Pretrain phase and get expert GMM list $X$          ▷ See algo. 2
2: Setup new ALP-GMM ($\rho_{rnd} = 0, \mathcal{T}$)          ▷ See algo. 1
3: Setup IN          ▷ See algo. 3
4: **loop** Stop after sampling $E$ tasks (including pre-train)
5:      Get composite GMM $G$ from the current GMM of both ALP-GMM and IN
6:      $\rho_{low}\%$ of the time, sample a random parameter $\tau \in \mathcal{T}$
7:      Else, sample $\tau$ from a Gaussian chosen proportionally to its $LP$
8:      Generate env. with $\tau$, send it to student $s_\theta$ and observe episodic reward $r_\tau$
9:      Send $(p, r_p)$ pair to both ALP-GMM and IN
10: Add student's training data to $\mathcal{H}^S$
11: **Return** $s_\theta$

---

## 5 Experiments and Results

We organize the analysis of our proposed Meta-ACL algorithm around 3 experimental questions:

- What are the properties and important components of AGAIN (sec. 5.1)? In this section, we will leverage a toy environment without DRL students to conduct systematic experiments.

- Does AGAIN scale well to Meta-ACL scenarios with DRL students (sec. 5.2)? Here, we will present a new Box2D locomotion environment that will be used to conduct our experiments.

- Can AGAIN be used for single learners (sec. 5.3)? Here we will show that it can be useful to derive curriculum priors even for a single student (i.e. without any teaching history $\mathcal{H}^S$).

**Considered baselines and AGAIN variants.** In the aforementioned experiments, we compare AGAIN to the following conditions:

- IN, an ablation, which directly use the expert curriculum instead of combining it with ALP-GMM

- AGAIN_RND, an ablation which do not perform pre-tests and instead directly extract curriculum priors from a randomly selected student in the teaching history $\mathcal{H}^S$.

- AGAIN_GT, an AGAIN variant with privileged information: instead of performing pre-tests, this condition is given access to the ground truth student distribution to condition from which previously trained student to extract curriculum priors.

- *Random*, an ACL condition, randomly sampling tasks for its student.

- ADR, an existing ACL condition from (OpenAI et al., 2019), used in section 5.1 and 5.2.

- *Oracle*, an expert-made ACL approach, used in 5.3.

See appendix B for details on these conditions.

### 5.1 Analyzing Meta-ACL in a Toy Environment

To provide in-depth experiments on AGAIN, we first emancipate from DRL students through the use of a toy testbed. The objective of this environment is to simulate the learning of a student within a 2D task-encoding parameter space $\mathcal{T} = [0, 1]^2$, a.k.a. task space. This fake task space is uniformly divided in 400 square cells

$C \subset \mathcal{T}$, and each task $\tau \in \mathcal{T}$ sampled by the teacher is directly mapped to an episodic reward $r_\tau$ based on sampling history and whether $C$ is considered "locked" or "unlocked". Three rules enforce reward collection in $\mathcal{T}$:

1. Every cell $C$ starts "locked", except a randomly chosen one that is "unlocked".

2. If $C$ is "unlocked" and $\tau \in C$, then $r_\tau = min(|C|, 100)$, with $|C|$ the cumulative number of tasks sampled within $C$ while being "unlocked" (if $C$ is "locked", then $r_\tau = 0$).

3. If $|C| >= 75$, adjacent cells become "unlocked".

Given these rules, one can model students with different curriculum needs by assigning them different initially unlocked cells, which itself models what is "easy to learn" initially for a given student, and from where it can expand. See figure 5 for illustrations of this toy environment. Note that our toy testbed implements a fake task space along with a simulation of competence improvement from fake policy learners over this space: there is no learning happening, apart from the teacher learning a task sampling policy. For instance, a cell in the task space (a subspace) can be seen as a group of task-encoding parameters creating bipedal locomotion challenges of similar difficulty profiles. Another example would be that task-encoding parameters correspond to the initial position of blocks in a robotic manipulation environment where blocks must be arranged into fixed position.

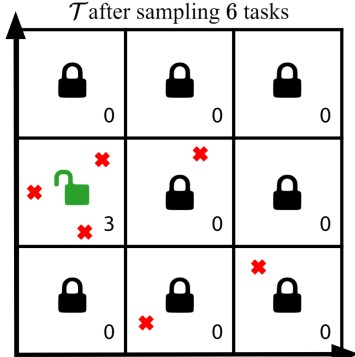 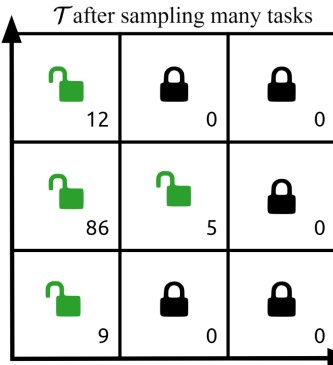

Figure 5: A toy environment to study teacher algorithms. **Left:** Teachers must focus their sampling strategy in unlocked areas ("learnable" cells) of the 2D task space. Red crosses illustrate this sampling procedure. Successful task sampling increments the competence counter of the cell. **Right:** If the teacher manages to focus its sampling strategy in unlocked cells, new cells will be unlocked, which simulates competence improvement.

### 5.1.1 Results

Instead of performing a pre-test to construct the $CP^{pre}$ vector of a student, we directly compute it by concatenating $|C|$ for all cells, giving a 400-dimensional $CP^{pre}$ vector. This vector is computed after 20k training episodes out of 200k. To study AGAIN, we first populate our history of trained students $\mathcal{H}^{\mathcal{S}}$ by training with ALP-GMM an initial classroom of 128 students drawn randomly from 4 fixed possible student types (i.e. 4 possible initially unlocked cell positions), and then test it on a new fixed set of 48 random students.

**Comparative analysis**  Figure 6 (left) showcases performance across training for our considered Meta-ACL conditions and ACL baselines. Both AGAIN and IN significantly outperform ALP-GMM ($p < .001$ for both, using Welch's t-test at 200k episodes). The initial performance advantage of IN w.r.t AGAIN is due to the greedy nature of IN, which only exploits the expert curriculum while AGAIN complements it with ALP-GMM for exploration. By the end of training, AGAIN outperforms IN ($p < .001$) thanks to its ability to emancipate from the curriculum priors it initially leverages. The regular CP-based curriculum priors selection used in AGAIN outperformed the random selection used in AGAIN_RND ($p < .001$ at 200k episodes), while being not significantly inferior to the Ground Truth variant AGAIN_GT ($p = 0.16$). Because we assume no expert

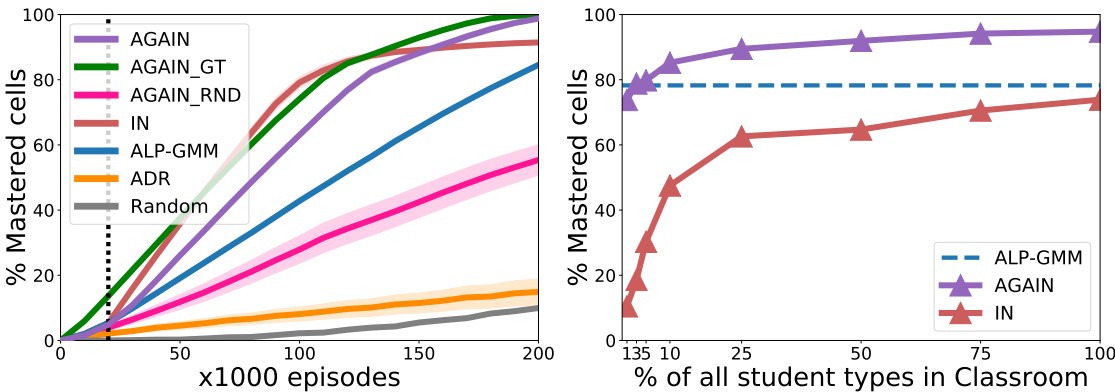

Figure 6: **Left:** By leveraging meta-learned curriculum priors w.r.t to its students, AGAIN outperforms regular ACL approaches. Avg. perfs. with *sem* (standard error of the mean) plotted, 48 seeds. The vertical dashed black line indicates when pre-training ends for Meta-ACL conditions. **Right:** Impact of classroom size and sparsity on Meta-ACL performances. Post-training (200k ep.) avg perfs. plotted, 96 seeds.

knowledge over the set of students to train, i.e. their respective initial learning subspace is unknown, ADR– which relies on being given an initial easy task – fails to train most students when given randomly selected starting subspace (among the 4 possible ones). By contrast, this showcases the ability of AGAIN to autonomously and efficiently infer such expert knowledge.

**Varying classroom size experiment** An important property that must be met by a meta-learning procedure is to have a monotonic increase of performance as the database of information being leveraged increases. Another important expected aspect of Meta-ACL is whether the approach is able to generalize curriculum generation to students that were never seen before. To assess whether these properties hold on AGAIN, we consider the full student distribution of the toy environment, i.e. 400 possible student types. We populate a new history $\mathcal{H}^{\mathcal{S}}$ by training (with ALP-GMM) a 400-students classroom (one per student type). We then analyze the end performance of AGAIN and IN on a fixed test set of 96 random students when given increasingly smaller subsets of $\mathcal{H}^{\mathcal{S}}$. The smaller the subset, the harder it becomes to generalize over new students. Results, shown in fig. 6 (right), demonstrate that both AGAIN and IN do have monotonic performance increasments as the classroom grows. With as little as 10% of possible students in the classroom, AGAIN statistically significantly ($p < .001$) outperforms ALP-GMM on the new student set, i.e. it generalizes to never seen before students.

## 5.2 Meta-ACL for DRL students in the Walker-Climber Environment

To study Meta-ACL with DRL students, we present *Walker-Climber*, a new Box2D locomotion environment (inspired from Romac et al. (2021)). *Walker-Climber* features a 2D parametric PCG that encodes a large space of tasks (see fig. 7). The first task parameter controls the spacing between walls that are positioned along the track, while the second task parameter sets the y-position of a gate that is added to each wall. Positive rewards are collected by going forward. To simulate a multi-modal distribution of students well suited to study Meta-ACL, we randomize the student's morphology for each new training (i.e. each seed): It can be embodied in either a bipedal walker, which will be prone to learn tasks with near-ground gate positions, or a two-armed climber, for which tasks with near-roof gate positions are easiest. We also randomize the student's limb sizes, which can vary from the length visible in fig. 7 to 50% shorter.

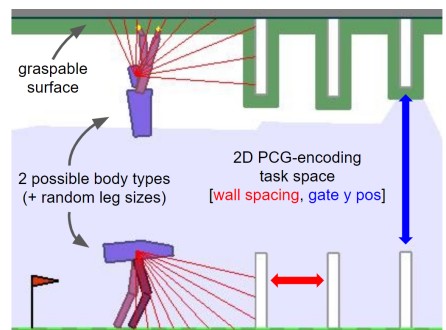

Figure 7: Our proposed *Walker-Climber* parametric env. to study Meta-ACL with DRL students.

### 5.2.1 Results

In the following experiments, our Meta-ACL variants leverage a teaching history $\mathcal{H}^\mathcal{S}$ built from a classroom of 128 randomly drawn Soft-Actor-Critic (Haarnoja et al., 2018) students (with varying embodiments and initial policy weights) trained with ALP-GMM. We compare ACL and Meta-ACL variants on a fixed set of 64 new students and report the mean percentage of mastered environments (i.e. $r > 230$) from 2 fixed expert test sets (one per embodiment type) across training. The $CP^{pre}$ vector is built using a uniform pre-test set of $m = 225$ tasks, performed after 2M agent steps out of 10. See appendix D for additional experimental details.

**Qualitative view** Figure 8 (bottom left) showcases the evolution of task sampling when using AGAIN to train a new student. Three distinct phases emerge along training: 1) A pre-training exploratory phase used to gather information about the student's capabilities, 2) After building the $CP^{pre}$ vector and inferring the most appropriate curriculum priors from $\mathcal{H}^\mathcal{S}$, AGAIN paces through the resulting IN curriculum while mixing it to ALP-GMM, and 3) AGAIN emancipates from IN after completing it.

**Comparative analysis** As shown in figure 8 (bottom right), through its use of curriculum priors, AGAIN outperforms the teaching performances of ALP-GMM on our environment. AGAIN's students master an average of 41% of the test set at 10M steps against 31% for ALP-GMM's ($p < .001$) after 10.5M steps (0.5M training steps added to account for AGAIN additional pre-test time). AGAIN performs better than its AGAIN_RND random prior selection variant, and is not statistically different ($p = 0.8$) from ground truth sampling (AGAIN_GT), although only by the end of training. While AGAIN and IN initially have comparable performances, after 7M training steps – a point at which most students trained with IN or AGAIN reached the last Gaussian mixture from IN– AGAIN starts to outperform IN, with a significant post-training advantage ($p < 0.02$). This showcases the advantage of emancipating from the expert curriculum once completed. As in the toy environment experiments, when given randomly selected starting subspaces (since we assume no expert knowledge), ADR fails to train most students. Figure 8 (top) showcases which tasks of the post-training test set were mastered for each SAC student trained with Random, ALP-GMM or AGAIN. AGAIN better exploits the learning capacities of its students, leading to a superior overall mastery of the task space.

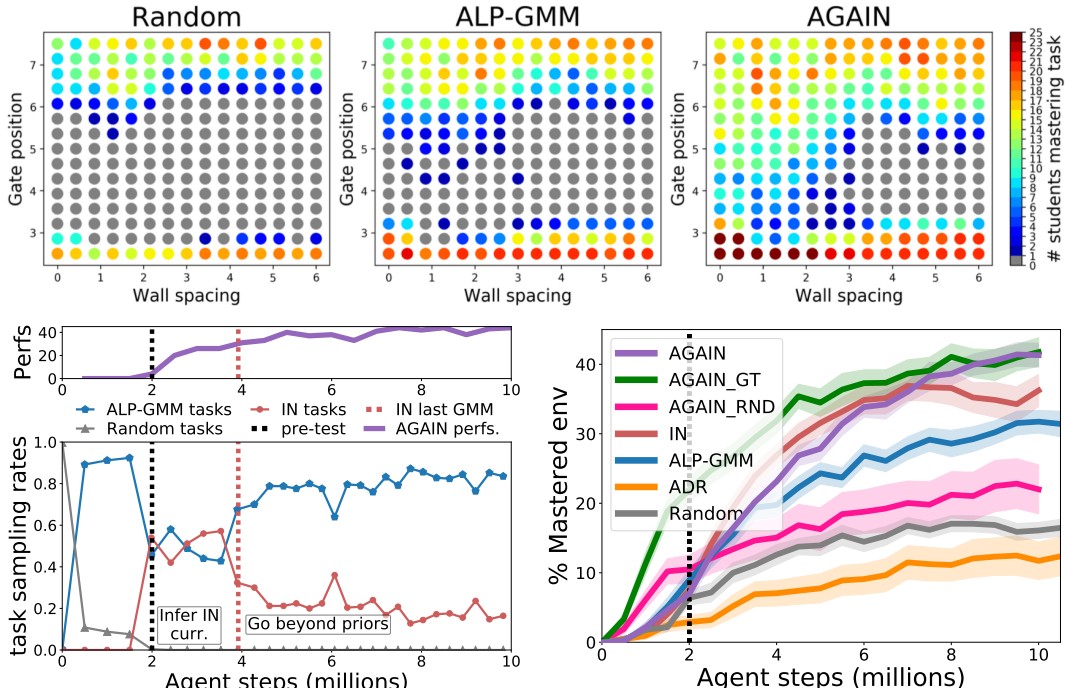

Figure 8: **Bottom Left:** Example run of evolution of task sampling with AGAIN in *Walker-Climber*. **Bottom Right:** Average performances of AGAIN with variants and baselines (same env). 64 seeds, sem plotted). The vertical dashed black line indicates when pre-training ends for Meta-ACL conditions. **Top:** Overall post-training performances for each student training (Random, ALP-GMM, AGAIN). Each test task (dot) is colored according to how many students (out of 64) mastered it (i.e. obtained $r > 230$).

### 5.3 Applying Meta-ACL to a Single Student: Trying AGAIN instead of Trying Longer

Given a single DRL student to train (i.e. no history $\mathcal{H}^{\mathcal{S}}$) and an expert knowledge-free setup, current ACL approaches leverage task-exploration (as in ALP-GMM). We hypothesize that these additional tasks presented to the DRL learner could have a cluttering effect on the gathered training data, i.e. it adds noise in its already brittle gradient-based optimization and leads to suboptimal performances. We propose to address this problem by modifying AGAIN to fit this no-history setup. To do so, we assume the ability to restart the student once along training, i.e. we drop the non-resetable assumption from classical ACL setups (section 3). More precisely, instead of pre-testing the student to find appropriate curriculum priors from $\mathcal{H}^{\mathcal{S}}$, we split the training of the target student into a two-stage approach, where 1) the DRL student is first trained with ALP-GMM (with high-exploration), and then 2) we extract curriculum priors from the training history of the first run and use them to re-train the same agent *from scratch*.

#### 5.3.1 Results

We test our modified AGAIN along with variants and baselines on the *Stump Tracks* environment proposed in Portelas et al. (2019), which generates walking tracks paved with stumps whose height and spacing are defined by a PCG-encoding 2D vector. Following Portelas et al. (2019), we test our approaches with both the default walker and a modified short-legged walker, which constitutes an even more challenging scenario (as the task space is unchanged). Performance is measured by tracking the percentage of mastered tasks from a fixed test set. See app. E for additional results. We use the same LP threshold to discard low-LP Gaussians when extracting curriculum priors as in our Walker-Climber experiments (i.e. $\delta_{LP} = 0.2$).

Figure 9 showcases our proposed approach on the short (top) and default (bottom) walker setups, with a SAC student (Haarnoja et al., 2018). In both cases, AGAIN statistically significantly outperform ALP-GMM ($p < 0.05$). This performance gap is most striking in the short walker setup. This result is expected: this hard training scenario is more likely to benefit from adaptive curriculum generation since there are less feasible task subspaces w.r.t. the default walker setup.

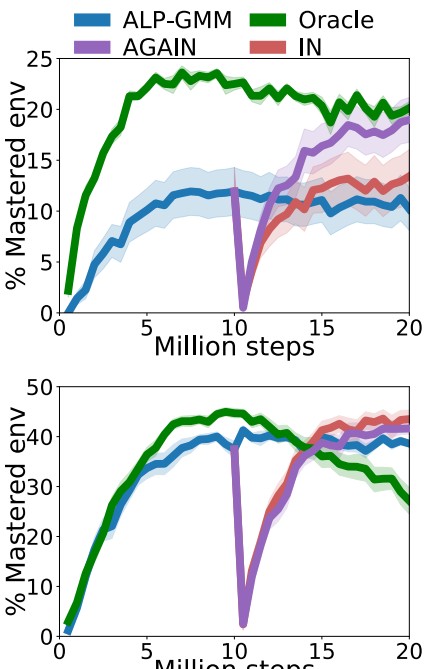

Figure 9: Given a single DRL student to train, AGAIN outperforms ALP-GMM in a parametric BipedalWalker environment. sem plotted, 32 seeds. **Top:** Experiments with *short* bipedal walkers. **Bottom:** Experiments with *default* bipedal walkers.

**AGAIN vs IN** In the default walker experiments, AGAIN and IN reach similar end-performances ($p > 0.05$). This is unsurprising: in this simple setting preliminary trainings on 10 million environment steps with ALP-GMM always (in our 32 seeds) find feasible task subspaces to focus on. This means appropriate curriculum priors can be consistently extracted for re-training, i.e. IN curricula are sufficient, and complementing it with exploration as in AGAIN is unnecessary. However, on the short walker scenario, mixing ALP-GMM with IN is essential: while IN end performances are not statistically significantly superior to ALP-GMM, AGAIN clearly outperforms ALP-GMM ($p < 0.01$), reaching a mean end performance of 19.0. This is due to the difficulty of the short walker scenario: after the preliminary 10 million training steps, 16/32 SAC students did not manage to learn any locomotion policy. All these run failures led to many GMMs lists $\mathcal{X}$ used in IN to be of very low-quality, i.e. low-quality curriculum priors, which illustrates the advantage of AGAIN that is able to complement them with further exploration.

## 6 Conclusion and Discussion

In this work we attempted to motivate and formalize the study of Classroom Teaching problems, in which a set of diverse students have to be trained optimally, and we proposed to attain this goal through the use of Meta-ACL algorithms. We then presented AGAIN, a first Meta-ACL baseline, and demonstrated its advantages over classical ACL and variants for CT problems in both a toy environment and in *Walker-Climber*, a new parametric locomotion environment with DRL learners. We also showed how AGAIN can bring performance gains over ACL in classical single student ACL scenarios.

**Limitations & future work.** AGAIN is a first Meta-ACL *baseline*, i.e. a first step aiming to seed further research. Many parts of its learning pipeline could be improved. For instance, in future work, instead of building large pre-test sets spanning over the task space, AGAIN could use adaptive approaches to build compact pre-test sets, e.g. using decision tree-based test pruning methods. AGAIN relies on pre-defining the length of the initial pre-training period. This hyperparameter is crucial and must be carefully selected by the experimenter: if pre-training is too short, the pre-tests and resulting competence profiles of students will not be easily separable. If too long, pre-training will drain training time for the main training session, thus hindering performances. An interesting avenue for future work would be to study how to avoid relying on pre-tests to select curriculum priors, e.g. to extract similarity measures between students based on their training history $\mathcal{H}^{int}$. Additionally, although for simplicity we focused on extracting useful curriculum priors from a single student in the history of previously trained students, combining curriculum priors from multiple previously trained learners, or even adaptively switching from which student to extract curriculum priors along training, appears like interesting research directions. Our current experiments are centered around Box2D locomotion environments: to strengthen the validity and applicability of Meta-ACL approaches, additional experiments on other domains (e.g. robotic manipulation, pixel-based environments) constitute a valid endeavor for future work.

While AGAIN is built on top of an existing ACL algorithm, developing an end-to-end Meta-ACL algorithm that generates curricula using a DRL teacher-policy trained across multiple students is also a promising line of work to follow. In practice, approaching this task-level control problem with classical DRL algorithms is challenging because of sample efficiency: an ACL policy has to be learned and exploited along interaction windows typically around a few tens of thousands of steps. This has to be compared to the tens of millions or sometimes billions of interaction steps necessary to train a DRL policy for robotic control tasks. For this reason, most recent ACL research has focused on reducing the teaching problem into a Multi Armed Bandit setup, which ignores the sequential dependency over student states implied in POMDP settings (Matiisen et al., 2017; Mysore et al., 2018; Colas et al., 2019; Portelas et al., 2019). One potential research direction towards this end-to-end Meta-ACL goal would be to study how to modify the DQN curriculum generator proposed in Narvekar & Stone (2020) to fit a Classroom Teaching scenario.

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

# A  AGAIN

**IN variants.**  In order to filter the list $X_{raw}$ (see eq. 5) of GMMs extracted from a previous student from $\mathcal{H}^S$ trained with ALP-GMM into $X$ and use it as an expert curriculum, we remove any Gaussian with a $LP_{ti}$ below $\delta_{LP} = 0.2$ (the LP dimension is normalized between 0 and 1, which requires the experimenter to choose an approximate potential reward range, set to $[-150, 350]$ for all experiments on Box2D locomotion environments (sec. 5.2 and sec. 5.3). When all Gaussians of a GMM are discarded, the GMM is removed from $X$. In practice, it allows to 1) remove non-informative GMMs corresponding to the initial exploration phase of ALP-GMM, when the learner has not made any progress (hence no LP detected by the teacher), and 2) discard $X_{raw}$ entirely and its associated student $s$ from the teaching history if ALP-GMM never detected high-LP Gaussians, i.e. it failed to train student $s$. We propose 3 variants to iterate over $X$ to generate a task-sampling curriculum:

- *Pool-based (IN-P), algo. 5:* A rather crude approach is to disregard the ordering of $X$ and merge the trajectory of GMMs into a single large GMM. This GMM is used for every task sampling step, i.e. the curriculum is fixed.

- *Time-based (IN-T), algo. 6:* In this version $X$ is stepped in periodically at a rate $N$, which we set to 250 (same as the fitting rate of ALP-GMM).

- *Reward-based (IN-R), algo. 3:* A more adaptive option is to iterate over $X$ only once the mean episodic reward over tasks recently sampled from the current GMM matches or surpasses the mean episodic reward recorded during the initial ALP-GMM run (on the same GMM). The IN-R approach requires extracting additional data from the first run, in the form of a list $\mathcal{R}_{raw}$:

$$\mathcal{R}_{raw} = \{\mu_r^1, ..., \mu_r^t, \mu_r^T\} \ \ s.t \ \ |\mathcal{R}_{raw}| = |X_{raw}|, \tag{7}$$

  with T the total number of GMMs in the first run (same as in $X_{raw}$), and $\mu_r^t$ the mean episodic reward obtained by the first DRL agent during the last 50 tasks sampled from the $t^{th}$ GMM. $\mathcal{R}$ is simply obtained by removing any $\mu_r^t$ that corresponds to a GMM discarded while extracting $X$ from $X_{raw}$. The remaining rewards are then used as thresholds in IN-R to decide when to switch to the next GMM in $X$.

Regardless of the selection process, given a GMM, a new task is selected by sampling its parameters from a Gaussian selected proportionally to its $LP_{ti}$ value.

**AGAIN variants**  The AGAIN approach (see algo. 4) use both IN (R,T or P) and ALP-GMM (without the random bootstrapping period) for curriculum generation. Our main experiments use IN-R as it is the highest performing variant (see app. C). This means that in the main body of this article, AGAIN = AGAIN-R and IN = IN-R.

**AGAIN details and hyperparameter choices**  While combining ALP-GMM to IN, we reduce the residual random sampling of ALP-GMM from $\rho_{high} = 10\%$, used for the pretrain phase, to either $\rho_{low} = 2\%$ for experiments presented in sec. 5.1 and sec. 5.3, or $\rho_{low} = 0\%$ for experiments done in the *Walker-Climber* environment in sec. 5.2 (here we found $\rho_{low} = 0\%$ to be beneficial in terms of performances w.r.t. $\rho_{low} = 2\%$, which means that the task-exploration induced by the periodic GMM fit of ALP-GMM was sufficient for exploration). In AGAIN-R and AGAIN-T, when the last GMM $p(T)$ of the IN curriculum is reached, we switch the fixed $LP_{Ti}$ values of all IN Gaussians to periodically updated LP estimates, i.e. we allow AGAIN to modulate the importance of $p(T)$ for task sampling depending on its current student's performance. Regarding $\rho_{high}$, i.e. the percentage of random task sampling when using ALP-GMM for pre-training, we initially used 20% as in the ALP-GMM paper. We then tried lowering it to 10%, and observed slightly better performances without exploration issues, and therefore used it for all of our experiments. In the AGAIN approach, $\delta_{LP}$ is a normalized LP value that can go from 0 (all Gaussians accepted) to 1 (all Gaussians rejected). We set its value to 0.2 as a way to reject any zero-LP Gaussians (or low-LP Gaussians), even if LP estimates for such Gaussians are noisy and go slightly over 0. Several values between 0.05 and 0.3 were tested (on former

AGAIN versions): no significant differences were observed. Regarding the pre-training budget in AGAIN: this hyperparameter must be set such that students pre-train long-enough to acquire reasonably separable competence profiles. In our Walker-Climber experiments we set it to 20% of the training budget (2M episodes) based on both reported performance curves from (Portelas et al., 2019) in related Box2D setups and from initial experiments with classical ACL teachers in the Walker-Climber environment. We also tried 10% of the training budget, which led to less separable competence profiles. As a whole, we did not experience many difficulties while setting up these hyperparameters, which is encouraging regarding the transfer potential to new domains.

---

**Algorithm 5**   Inferred progress Niches - Pool-based (IN-P)

---

**Require:** Student policy $s_\theta$, teacher training history $\mathcal{H}^\mathcal{S}$, task-encoding parameter space $\mathcal{T}$, LP threshold $\delta_{LP}$, update rate $N$, experimental budget $E$, experimental pre-train budget $E_{pre}$, pre-test set size $m$, number of neighbors for student selection $k$, random sampling ratio $\rho_{high}$

1: Launch Pretrain phase and get expert GMM list $X$                                # See algo. 2
2: Initialize pool GMM $G_{IN}$, containing all Gaussians from $X$
3: **loop** Stop after sampling $E$ tasks (including pre-train)
4:      Sample $\tau$ from a Gaussian in $G_{IN}$ chosen proportionally to its $LP_{ti}$
5:      Generate env. with $\tau$, send it to student $s_\theta$
6: Add student's training data to $\mathcal{H}^\mathcal{S}$
7: **Return** $s_\theta$

---

**Algorithm 6**   Inferred progress Niches - Time-based (IN-T)

---

**Require:** Student policy $s_\theta$, teacher training history $\mathcal{H}^\mathcal{S}$, task-encoding parameter space $\mathcal{T}$, LP threshold $\delta_{LP}$, update rate $N$, experimental budget $E$, experimental pre-train budget $E_{pre}$, pre-test set size $m$, number of neighbors for student selection $k$, random sampling ratio $\rho_{high}$

1: Launch Pretrain phase and get expert GMM list $X$                                # See algo. 2
2: Initialize expert curriculum index $i_c$ to 0
3: **loop** Stop after sampling $E$ tasks (including pre-train)
4:      Set $i_c$ to $min(i_c + 1, len(X))$
5:      Set current GMM $G_{IN}$ to $i_c^{th}$ GMM in $X$
6:      **loop** $N$ times
7:          Sample $\tau$ from a Gaussian in $G_{IN}$ chosen proportionally to its $LP_{ti}$
8:          Send $E(a \sim \mathcal{A}(p))$ to student $s_\theta$
9: Add student's training data to $\mathcal{H}^\mathcal{S}$
10: **Return** $s_\theta$

---

# B Considered ACL and Meta-ACL Teachers

**Meta-ACL variants** Our proposed approach, AGAIN, is based on the combination of an inferred expert curriculum with ALP-GMM, an exploratory ACL approach. In appendix A, we present 3 approaches to use such an expert curriculum, giving the AGAIN-R, AGAIN-P and AGAIN-T algorithms (only AGAIN-R is used in our main experiments). In our experiments, we also consider ablations where we only use the expert curriculum, giving the IN-R, IN-P and IN-T variants. We also consider two additional AGAIN variants that do not use our proposed CP-based student selection method:

- AGAIN with Random curriculum prior selection (AGAIN_RND), a lower-baseline which do not perform pre-tests and instead directly extract curriculum priors from a randomly selected student in the teaching history $\mathcal{H}^{\mathcal{S}}$.

- AGAIN with Ground Truth selection (AGAIN_GT), an upper-baseline using privileged information. Instead of performing the knn algorithm in the CP space (see section 4), this approach directly uses the true student distribution. For instance, in the *Walker-Climber* environment, given a new student $s$, AGAIN_GT selects the $k$ previously trained students from $\mathcal{H}^{\mathcal{S}}$ that are morphologically closest to $s$ (i.e. same embodiment type and closest limb sizes), and extracts curriculum priors $X$ from the student with the highest score $j_s$ (see sec. 4).

Note that both for AGAIN_RND and AGAIN_GT, there is no need to pre-test the student, which means we can use the IN expert curriculum directly at the beginning of training rather than after a pre-training phase.

**ACL conditions** A first natural ACL approach to compare our AGAIN variants to is ALP-GMM, the underlying ACL algorithm in AGAIN. We also add as a lower-baseline a random curriculum teacher (Random), which samples tasks' parameters randomly over the task space.

In both the toy environment (sec. 5.1, toy env. for short) and the *Walker-Climber* environment (sec. 5.2), we additionally compare to Adaptive Domain Randomization (ADR), an ACL algorithm proposed in OpenAI et al. (2019), which is based on inflating a task distribution sampling from a predefined initially feasible task $\tau_{easy}$ (w.r.t a given student). Each lower and upper boundaries of each dimension of the sampling distribution are modified independently with step size $\Delta_{step}$ whenever a predefined mean reward threshold $r_{thr}$ is surpassed over a window (of size $q$) of tasks occasionally sampled (with probability $\rho_b$) at the sampling dimension boundary. More details can be found in OpenAI et al. (2019). In our experiments, as we do not assume access to expert knowledge over students sampled within the student distribution, we randomize the setting of $\tau_{easy}$ uniformly over the task space in *Walker-Climber* experiments and uniformly over the 4 possible student starting subspaces in toy env. experiments. Based on the hyperparameters proposed in OpenAI et al. (2019) and on informal hyperparameter search, we use $[\rho_b = 0.5, r_{thr} = 1, \Delta_{step} = 0.05, q = 10]$ in toy env. experiments and $[\rho_b = 0.5, r_{thr} = 230, \Delta_{step} = 0.1, q = 20]$ in *Walker-Climber* experiments.

In experiments described in sec 5.3, we compare our approaches to an oracle condition (Oracle), which is a hand-made curriculum that is very similar to IN-R, except that the list $X$ is built using expert knowledge before training starts (i.e. no pre-train and pre-test phases), and all reward thresholds $\mu_r^i$ in $\mathcal{R}$ (see eq. 7) are set to 230, which is an episodic reward value often used in the literature as characterizing a default walker having a "reasonably efficient" walking gate in environments derived from the Box2D gym environment BipedalWalker, e.g. in Wang et al. (2019) or Portelas et al. (2019). In practice, Oracle starts proposing tasks from a Gaussian (with std of 0.05) located at the simplest subspace of the task space (i.e. low stump height and high stump spacing) and then gradually moves the Gaussian towards the hardest subspaces (high stump height and low stump spacing) by small increments (50 steps overall) happening whenever the mean episodic reward of the DRL agent over the last 50 proposed tasks is superior to 230.

# C Analyzing Meta-ACL in a Toy Environment

In this section, we report the full comparative experiments done in the toy environment, which includes comparisons with AGAIN-T and AGAIN-P to AGAIN-R, shown in table 1. We also provide visualizations of

the CP-based curriculum priors selection process (see fig. 11) happening after the pretraining phase in AGAIN along with a visualization of the fixed set of 96 randomly drawn students used to perform the varying classroom experiments reported in sec. 5.1 (see fig. 10).

**Additional comparative analysis** Table 1 summarizes the post-training performances obtained by our considered Meta-ACL conditions and ACL baselines on the toy environment on a fixed student test-set of 48 randomly drawn students (among 4 possible student types). Meta-ACL conditions are given a teaching history $\mathcal{H}^{\mathcal{S}}$ created by training an initial classroom of 128 students. Using a Reward-based iterating scheme over the inferred expert curriculum (AGAIN-R and IN-R) outperforms the Time-based and Pool-based variants ($p < .001$). This result was expected as both these last two variants do not have flexible mechanisms to adapt to the student being trained. The pool based variants (AGAIN-P and IN-P), which discard the temporal ordering of the expert curriculum are the worst performing variants, statistically significantly inferior to both Reward-based and Time-based conditions ($p < .001$).

Table 1: **Experiments on the toy environment.** The average performance with standard deviation after 200k episodes is reported (48 seeds per conditions). For Meta-ACL variants we report results with column 1) the regular CP-based curriculum prior selection performed after 20k pre-training episodes, column 2) An ablation that performs the selection at random before training, and column 3) An oracle condition selecting before training the curriculum prior using student ground truth type. * Denotes stat. significant advantage w.r.t. ALP-GMM (Welch's t-test at $200k$ ep. with $p < 0.05$).

| Condition | Regular | Random | Ground Truth |
|---|---|---|---|
| AGAIN-R | 98.8 +- 4.8* | 55.4 +- 32.2 | 99.8 +- 0.9* |
| IN-R | 91.4 +- 3.4* | 26.3 +- 41.1 | 92.5 +- 3.0* |
| AGAIN-T | 84.3 +- 3.8 | 38.6 +- 34.1 | 89.0 +- 1.7* |
| IN-T | 79.0 +- 12.0 | 30.3 +- 37.3 | 88.9 +- 1.7* |
| AGAIN-P | 38.2 +- 7.5 | 9.3 +- 9.2 | 14.8 +- 1.2 |
| IN-P | 40.6 +- 6.4 | 9.2 +- 9.0 | 15.1 +- 1.2 |
| ALP-GMM | 84.6 +- 3.4 | | |
| ADR | 14.9 +- 27.4 | | |
| Random | 10.0 +- 0.8 | | |

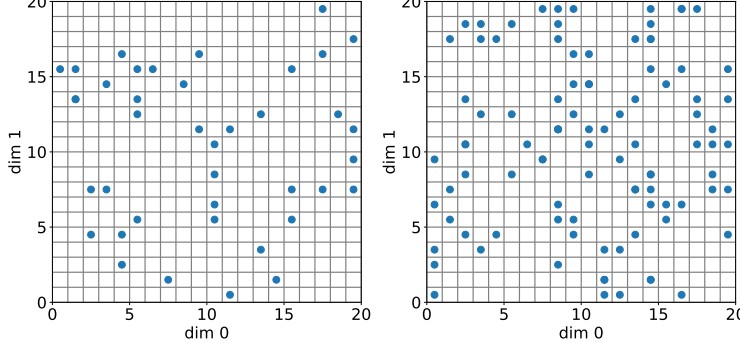

Figure 10: Additional visualizations for the varying classroom size experiments (sec. 5.1). **Left:** Visualization of the starting cells of students from a 10% sample of a classroom of 400 students (one per student type) trained with ALP-GMM and used to populate the teaching history $\mathcal{H}^{\mathcal{S}}$. Each blue circle marks the starting cell of each student (i.e. its type) within the 2D parameter space $\mathcal{T}$, which is an initial learning subspace that needs to be detected by the teacher for successful training. **Right:** Visualization of the fixed set of 96 randomly drawn students that have to be trained by Meta-ACL variants given $\mathcal{H}^{\mathcal{S}}$. As not all student types are represented in $\mathcal{H}^{\mathcal{S}}$, Meta-ACL approaches have to generalize their curriculum generation to these new students.

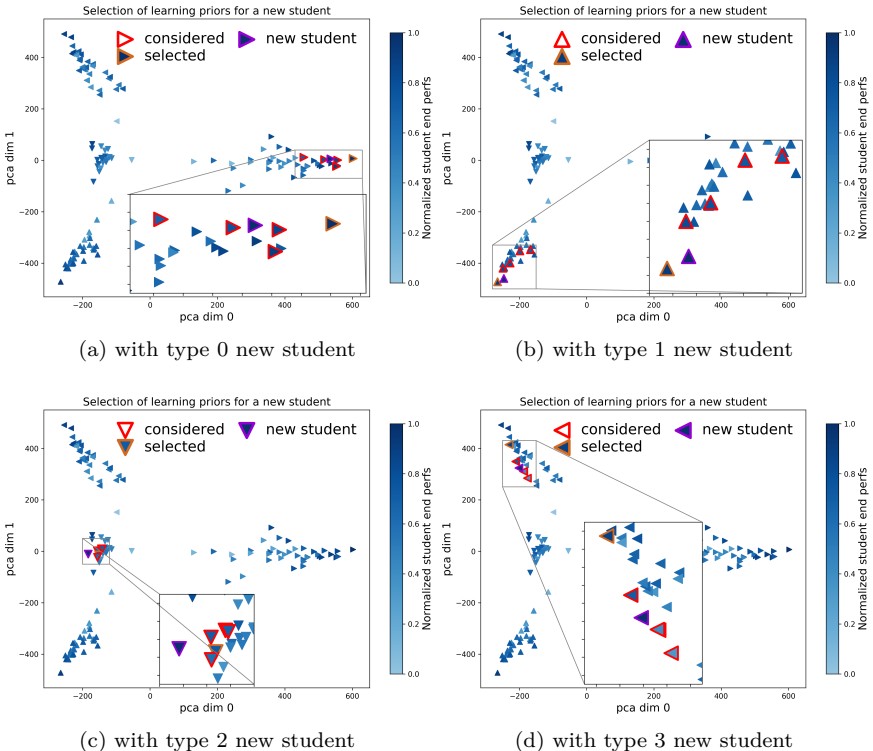

(a) with type 0 new student

(b) with type 1 new student

(c) with type 2 new student

(d) with type 3 new student

Figure 11: **Examples of student selection process in** 4**-student type toy environment.** In all figures, we plot the 2D PCA visualization of the $CP^{pre}$ vectors (after pre-training) of the initial classroom (128 students) trained with ALP-GMM and used to populate the teaching history $\mathcal{H}^{\mathcal{S}}$ used by AGAIN variants in our 4-student type toy env experiments (see sec. 5.1). We then use these 4 figures to showcase the selection process happening in 4 different AGAIN-R runs (one per student type). Each triangle represents a student, whose ground truth type (i.e. its initial learning cell) is denoted by the orientation of the triangle. Given a new student to train, AGAIN pretrains the student, constructs its CP vector (purple border triangle), infers the k closest previously trained students from $\mathcal{H}^{\mathcal{S}}$ (red and golden border triangles), and use the one with highest end of training performance (i.e. highest score $s$, see sec. 4), denoted by a golden border triangle, to infer curriculum priors for the new student.

# D  Meta-ACL for DRL Students in the *Walker-Climber* environment

In this section we give additional details on the *Walker-Climber* environment presented in section 5.2, and we provide additional details and visualizations on the experiments that were performed on it.

**Details on the *Walker-Climber* environment.**  In our experiments, we bound the wall spacing dimension of the task space to $\Delta_w = [0, 6]$, and the gate y position to $\mu_{gate} = [2.5, 7.5]$. In practice, a single task-encoding parameter tuple $(\mu_{gate}, \Delta_w)$ encodes a stochastic task , since for each new wall along the track we add an independent Gaussian noise to each wall's gate y position $\mu_{gate}$. Examples of *Walker-Climber* tasks randomly sampled within these bounds are available in figure 12 (right). At the beginning of training a given DRL policy, the agent is embodied in either a bipedal walker morphology with two joints per legs or a two-armed climber morphology with 3-joints per arms ended by a grasping "hand". Both morphologies are controlled by torque. Climbers have an additional action dimension $g \in [-1, 1]$ used to grasp: if $g \in [-1, 0[$, the climber closes its gripper, and if $g \in ]0, 1]$ it keeps it open. To avoid falling (which aborts the episode with a $-100$ penalty) while moving forward to collect rewards, climber agents must learn to swing themselves forward by successive grasp-and-release action sequences. To increase the diversity of the student distribution, we also randomize limb sizes. See figure 12 (left) for examples of randomly sampled embodiments.

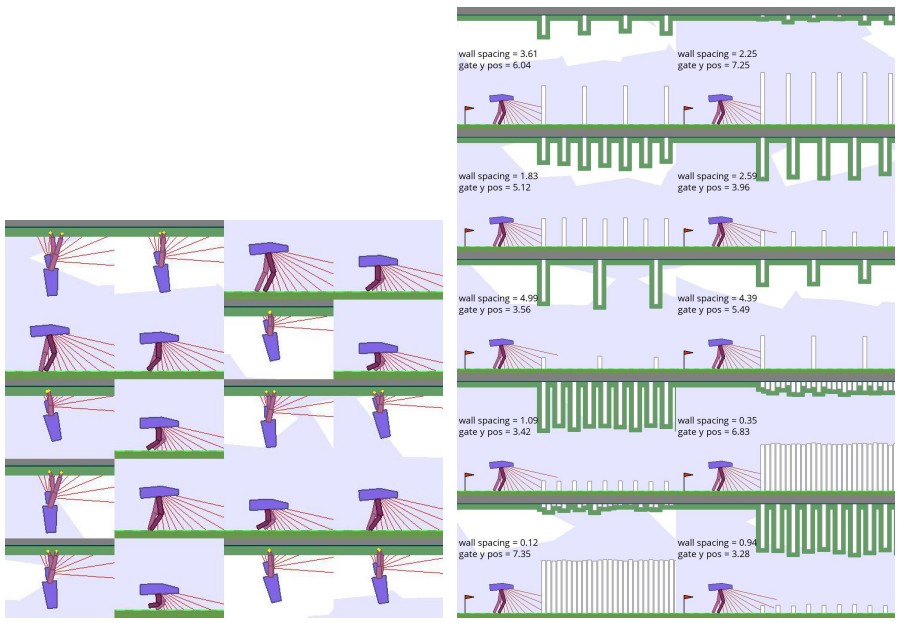

Figure 12: Visualizations of the student space and the task space of the *Walker-Climber* environment.**Left:** Examples of possible agent embodiments (randomly set for a given DRL learner before training starts). **Right:** Examples of randomly sampled *Walker-Climber* tracks.

**Soft Actor-Critic students**   In our experiments, we use an implementation of Soft Actor-Critic provided by OpenAI[1] (MIT license). We use a 2 layered (400,300) network for V, Q1, Q2 and the policy. Gradient steps are performed each 10 environment steps, with a learning rate of 0.001 and a batch size of 1000. The entropy coefficient is set to 0.005.

**Evaluation procedure**   To report the performance of our students on the *Walker-Climber* environment, we use two separate test sets, one per embodiment type. For walkers we use a 100-tasks test set, uniformly sampled over a subspace of the task space with $\Delta_w \in [0, 6]$ and $\mu_{gate} \in [2.5, 3.6]$, which loosely corresponds to walking tracks from the test set of our Stump Tracks environment from Portelas et al. (2019). For climbers, because there are no similar experiments in the literature , and since it is hard to infer beforehand what will be achievable by such a morphology, we simply use a uniform test set of 225 tasks sampled over the full task space. Importantly, the customized test set used for walkers is solely used for visualization purposes. In our AGAIN approaches, we pre-test all students with the expert-knowledge-free set of 225 tasks uniformly sampled over the task space.

**Compute resources**   Each of the 576 seeds required to reproduce our experiments (128 seeds for the classroom and $7 \times 64$ seeds for our 7 conditions) takes 36 hours on a single cpu. This amounts to around 21 000 cpu hours. Each run requires less than 1 GB of RAM.

**Visualizing student diversity.**   To assess whether our proposed multi-modal distribution of possible students in the *Walker-Climber* environment do have diverse competence profiles (which is desirable as it creates a challenging Meta-ACL scenario), we plot the 2D PCA of the post training CP vector for each students of the initial classroom trained with ALP-GMM (used to populate $\mathcal{H}^{\mathcal{S}}$). The result, visible in figure 13 (top), shows that climber-students and walker-students are located in two independent clusters, i.e. they do have clearly different competence profiles. The spread of each of the clusters also demonstrates that variations in initial policy parameters and limb sizes also creates students with diverse learning potentials. The competence differences between walkers and climbers can also be seen in Figure 13 (left and right),

---

[1]https://github.com/openai/spinningup

which shows the episodic reward obtained for each of the 225 tasks of the CP vector after training by a representative walker student (left) and climber student (right).

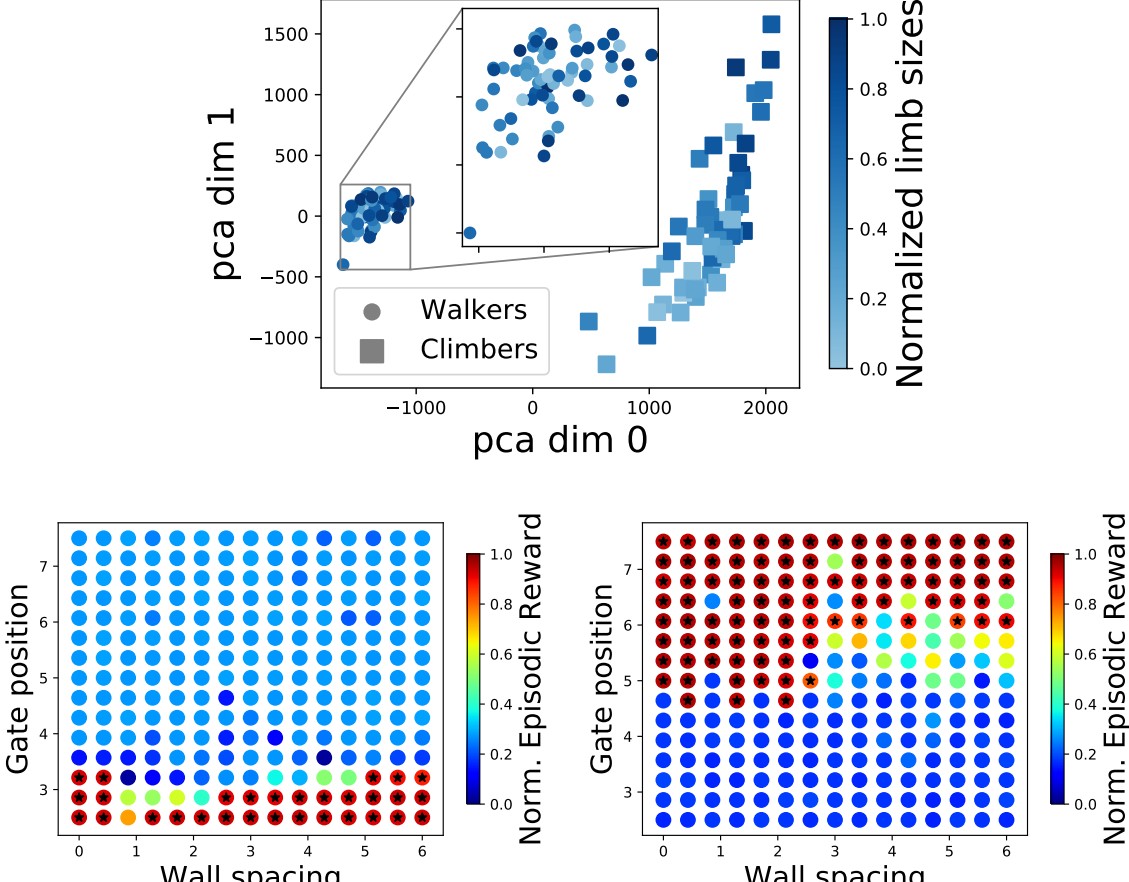

Figure 13: **top:** PCA of classroom's CP vector (128 students) after being trained for 10M student steps with ALP-GMM. **left and right:** Episodic reward obtained for each task that compose the CP vector by a walker-student (left) and a climber-student (right) of this classroom. Stars are added for all tasks for which the agent obtained more than $r = 230$ (which corresponds to an efficient locomotion policy). Walkers only manage to learn tasks with very low gate positions while climbers learn only tasks with medium to high gate positions.

# E    Applying Meta-ACL to a Single Student: Trying AGAIN instead of Trying Longer

In the following section we report all experiments on applying AGAIN variants to train a single DRL student (i.e. no history $\mathcal{H}^{\mathcal{S}}$), which is briefly presented in sec. 5.3.

**Parametric BipedalWalker env.**    We test our modified AGAIN variants along with baselines on the *Stump Tracks* environment from Portelas et al. (2019), which generates walking tracks paved with stumps whose height and spacing are defined by a 2D parameter vector used for the procedural generation of tasks. We keep the original bounds of this task space, i.e. we bound the stump-height dimension to $\mu_h \in [0, 3]$ and the stump-spacing dimension to $\delta_s \in [0, 6]$. As in Portelas et al. (2019), we also test our teachers when the learning agent is embodied in a modified short-legged walker, which constitutes an even more challenging scenario (as the task space is unchanged, i.e. more unfeasible tasks). The agent is rewarded for keeping its head straight and going forward and is penalized for torque usage. The episode is terminated after 1) reaching the end of the track, 2) reaching a maximal number of 2000 steps, or 3) head collision (for which the agent receives a strong penalty). See figure 14 for visualizations.

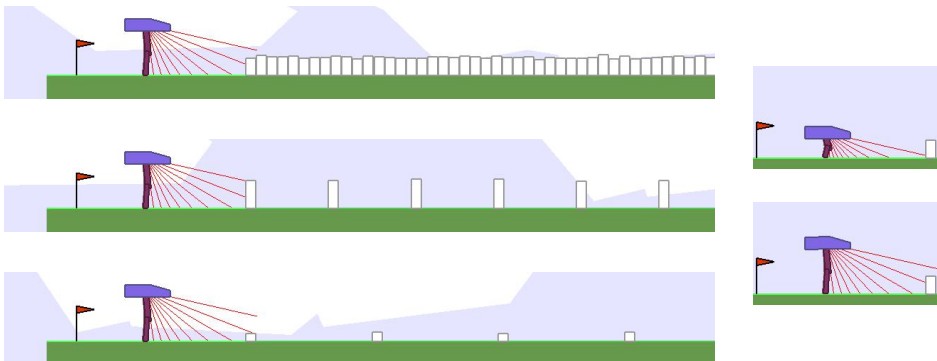

Figure 14: Parameterized BipedalWalker environment. **Left:** Examples of generated tracks. **Right:** The two walker morphologies tested on the environment.

**Results**    To perform our experiments, we ran each condition for either 10Millions (IN and AGAIN variants) or 20Millions (others) environment steps (30 repeats). The preliminary ALP-GMM runs used in IN and AGAIN variants correspond to the first 10 Million steps of the ALP-GMM condition (whose end-performance after 20 Million steps is reported in table 2. All teacher variants are tested when paired with a Soft-Actor Critic (Haarnoja et al., 2018) student, with same hyperparameters as in the *Walker-Climber* experiments (see app. D). Performance is measured by tracking the percentage of mastered tasks (i.e. $r > 230$) from a fixed test set of 100 tasks sampled uniformly over the task space. We thereafter report results for 2 independent experiments done with either default walkers or short walkers.

*Is re-training from scratch beneficial?* -  The end performances of all tested conditions are summarized in table 2 (performance curves are available in figures 15 and 16). Interestingly, retraining the DRL agent from scratch in the second run gave superior end performances than fine-tuning using the weights of the first run *in all tested variants*. This showcases the brittleness of gradient-based training and the difficulty of transfer learning. Despite this, even fine-tuned variants reached superior end-performances than classical ALP-GMM, meaning that the change in curriculum strategy in itself is already beneficial.

*Is it useful to re-use ALP-GMM in the second run?* -  In the default walker experiments, AGAIN-R, T and P conditions mixing ALP-GMM and IN in the second run reached lower mean performances than their respective IN variants. However, the exact opposite is observed for IN-R and IN-T variants in the short walker experiments. This can be explained by the difficulty of short walker experiments for ACL approaches, leading to 16/30 preliminary 10M steps long ALP-GMM runs to have a mean end-performance of 0, compared to 0/30 in the default walker experiments. All these run failures led to many GMMs lists $X$ used in IN to be of very low-quality, which illustrates the advantage of AGAIN that is able to emancipate from IN using ALP-GMM.

| Condition | Short walker | Default walker |
|---|---|---|
| AGAIN-R | $19.0 \pm 12.0^*$ | $41.6 \pm 6.3^*$ |
| AGAIN-R (fine-tune) | $11.4 \pm 12.9$ | $39.9 \pm 4.6$ |
| IN-R | $13.4 \pm 14.4$ | $43.5 \pm 9.6^*$ |
| IN-R (fine-tune) | $11.2 \pm 12.3$ | $40.8 \pm 5.6$ |
| AGAIN-T | $15.1 \pm 11.9$ | $40.6 \pm 11.5$ |
| AGAIN-T (fine-tune) | $11.4 \pm 11.8$ | $40.6 \pm 3.8^*$ |
| IN-T | $13.5 \pm 13.3$ | $43.5 \pm 6.1^*$ |
| IN-T (fine-tune) | $10.7 \pm 12.3$ | $40.3 \pm 7.6$ |
| AGAIN-P | $13.6 \pm 12.5$ | $41.9 \pm 5.1^*$ |
| AGAIN-P (fine-tune) | $11.1 \pm 12.0$ | $41.5 \pm 3.9^*$ |
| IN-P | $14.5 \pm 12.6$ | $\mathbf{44.3} \pm 3.5^*$ |
| IN-P (fine-tune) | $12.2 \pm 12.5$ | $41.1 \pm 3.8^*$ |
| ALP-GMM | $10.2 \pm 11.5$ | $38.6 \pm 3.5$ |
| Oracle | $\mathbf{20.1} \pm 3.4^*$ | $27.2 \pm 15.2^-$ |
| Random | $2.5 \pm 5.9^-$ | $20.9 \pm 11.0^-$ |

Table 2: **Experiments on Parametric Bipedal-Walker** The avg. perf. with std. deviation after 10 Millions steps (IN and AGAIN variants) or 20 Million steps (others) is reported (30 seeds). For IN and AGAIN we also test variants that do not retrain the weights of the policy used in the second run *from scratch* but rather *fine-tune* them from the preliminary run.$^{*/-}$ Indicates whether perf. difference with ALP-GMM is statistically significant ie. $p < 0.05$ in a post-training Welch's student t-test ($^*$ for performance advantage w.r.t ALP-GMM and $^-$ for perf. disadvantage).

*Highest-performing variants.* - Consistently with the precedent analysis, mixing ALP-GMM with IN in the second run is not essential in default walker experiments, as the best performing ACL approach is IN-P. This most likely suggests that the improved adaptability of the curriculum when using AGAIN is outbalanced by the added noise (due to the low task-exploration). However in the more complex short walker experiments, mixing ALP-GMM with IN is essential, especially for AGAIN-R, which substantially outperforms ALP-GMM and other AGAIN and IN variants (see fig. 9), reaching a mean end performance of 19.0. The difference in end-performance between AGAIN-R and Oracle, our hand-made expert using privileged information who obtained 20.1, is not statistically significant ($p = 0.6$).

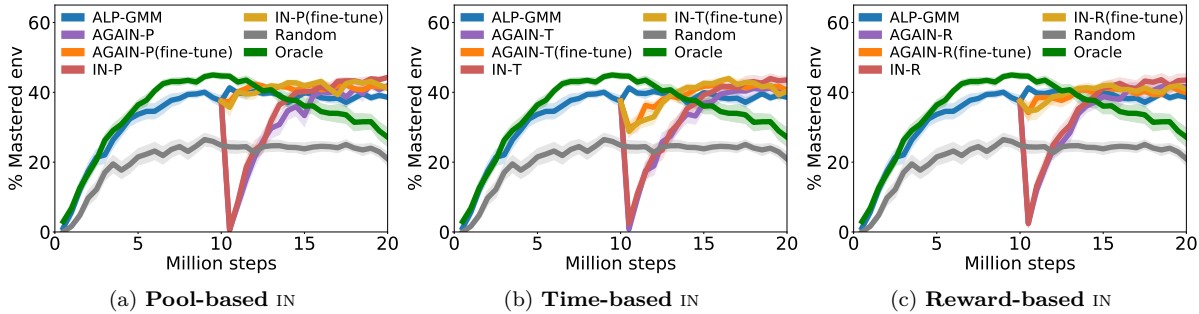

(a) **Pool-based** IN     (b) **Time-based** IN     (c) **Reward-based** IN

Figure 15: **Evolution of performance across 20M environment steps of each condition with default bipedal walker.** Each point in each curve corresponds to the mean performance (30 seeds), defined as the percentage of mastered tracks (ie. $r > 230$) on a fixed test set. Shaded areas represent the *sem.* Consistently with Portelas et al. (2019), which implements a similar approach, Oracle is prone to forgetting with default walkers due to the strong shift in task subspace (which is why it is not the best performing condition for default walker experiments).

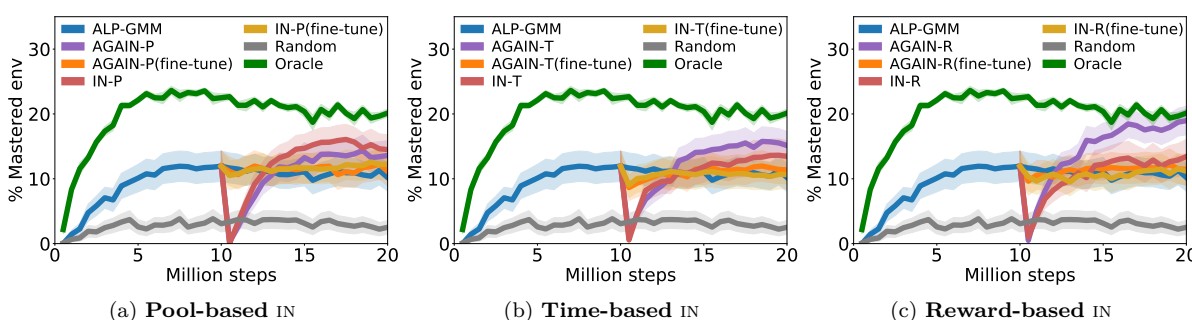

(a) **Pool-based** IN     (b) **Time-based** IN     (c) **Reward-based** IN

Figure 16: **Evolution of performance across 20M environment steps of each condition with short bipedal walker.** Each point in each curve corresponds to the mean performance (30 seeds), defined as the percentage of mastered tracks (ie. $r > 230$) on a fixed test set. Shaded areas represent the standard error of the mean.

