# OpenReview forum: "Meta Automatic Curriculum Learning for Classrooms of Deep RL Black-Box Students"
_TMLR — Rejected by TMLR_

### Review · Reviewer_RxN6 · 2022-10-03

**Summary Of Contributions:**

The paper extends automatic curriculum learning for deep reinforcement leanring to the multi-task setting where a task corresponds to a learner. The authors formalize this problem as an optimization problem, which consists in finding the meta-automatic curriculum learner such that the performance of the learner at the end of training is maximized. To solve this problem, they propose a heuristic method, which consists in pretraining the learner, finding the most similar and best-performing previous student s based on an evaluation of the pretrained learner on a fixed set of tasks, and guiding the pretrained learner using the training history of that student s. The method is evaluated in an artificial domain and a Box-2D environment.


**Broader Impact Concerns:**

There's no Broader Impact Statement section, however I think this is fine for this kind of paper.

**Requested Changes:**

I find the formalization could be more rigorous and could be improved (see also below), e.g.,
The notation "\mathcal D(\mathcal H_s^{int}) \rightarrow \tau" is not so great because it hides that the output of \mathcal D is a distribution over tasks. Moreover, Equation (1) doesn't use standard notations, and seems to suggest that the objective function is an expectation with respect to the task distribution. However, I guess that an expectation should also be taken over the distribution over training histories induced by \mathcal D. Similar remarks can be made about Equation (3). However, in the experimental part, the authors do average over several training (e.g., training histories).

Although the problem is formalized as an optimization problem, the proposed approach is actually a simple heuristic method that doesn't perform any optimization. The instantiated method depend on a certain number of design choices and hyperparameters. I guess the authors have tried different variations of their method. I would have liked them to be better justified and validated. For instance, how did you choose the values for the threshold \delta_{LP}, for \rho_{high} when combining ALP-GMM with IN, or for the pre-training budget? What values were tried? How sensitive are the results to those results? Since running the experiments is costly, it would not be practical to do that for a new a domain if the results are highly dependent on the choices of those hyperparameters.

In the single student case, I believe that the authors didn't explain how they extract curriculum priors from the training history of the first run. To fully understand and assess this part, I think this information should be provided.

**Strengths And Weaknesses:**

PROS
- New extension of automatic curriculum learning
- Simple heuristic to tackle this problem

CONS
- Experimental evaluation is a bit limited
- Writing and formalization could be more rigorous

---

> ### Author Response · Authors · 2022-10-07
> **Answer to Reviewer RxN6 (1/2)**
>
> We thank the reviewer for their valuable comments and suggestions. We were pleased to see that they saw our work as novel, and that they appreciated the conceptual simplicity of the AGAIN approach.
>
> RRxN6 pointed at several limitations of our current formulation, and suggested adding experimental details. In the following paragraphs, we present how we addressed these valid concerns:
>
> **A** - *R: “The notation $\mathcal D(\mathcal H_s^{int}) \rightarrow \tau$ is not so great because it hides that the output of $\mathcal{D}$ is a distribution over tasks”*
>
> **B** - *R: “Equation (1) doesn't use standard notations, and seems to suggest that the objective function is an expectation with respect to the task distribution. However, I guess that an expectation should also be taken over the distribution over training histories induced by $\mathcal{D}$. Similar remarks can be made about Equation (3). However, in the experimental part, the authors do average over several training (e.g., training histories).”*
>
> **C** - *R: “Although the problem is formalized as an optimization problem, the proposed approach is actually a simple heuristic method that doesn't perform any optimization.”*
>
> Thanks for pointing out limitations in our formalization. Following your comments, we performed the following modifications:
>
> **B & C** We do agree that formulating ACL and Meta-ACL as an optimization of post-training competence is not ideal. ACL methods in the literature predominantly use proxy objectives (e.g. learning progress, intermediate difficulty, adversarial reward maximization). We updated the manuscript to consider equations 1 and 2 as theoretical evaluation measures, not objectives. This allows us to write the ACL evaluation measure as $\mathcal{C} = \int_{\mathcal{T}} c^{E}_{\tau} \mathrm{d}\tau$
>
> and the Meta-ACL evaluation measure as $\hat{\mathcal{C}} = \int_{ \mathcal{S}}\int_{\mathcal{T}} c^E_{s,\tau} ~\mathrm{d}s\mathrm{d}\tau.$.From this perspective (a theoretical evaluation measure), our choice of integrals over the task space (and student-space in Meta-ACL) should appear more logical. This new formulation allows us to drop our former atypical notations. Note that we also spotted and removed a mistake that might have been a source of confusion: integrals are applied over $\mathcal{T}$, not $\tau \sim \mathcal{T}$, since they represent a theoretical evaluation over the full space (no sampling involved). The same fix was applied for the integral over the student space in the Meta-ACL evaluation measure.
>
> **A** We do agree that the current formulation of $\mathcal{D}$ is inadequate. We believe the source of confusion is that we call it a function, although it is mathematically not the case. We propose to reformulate it as a distribution: an ACL algorithm can be formalized as an algorithm producing a distribution $\mathcal{D}(\tau | \mathcal{H}^{int}_{s})$
>
> which sequentially samples (parameterized) tasks for $s$ given its training history $\mathcal{H}^{int}_{s}$.
>
> We perform the same modification for Meta-ACL: a Meta-ACL algorithm can be formulated as producing the following distribution:
>
> $$
> \hat{\mathcal{D}}(\tau \~|\~ \mathcal{H}^{int}_{s^{K}}, X) \~\~s.t.\~\~ X = f(\mathcal{H}^\mathcal{S})
> $$
> with H^S = [H^int_s0, H^int_s1, ..., H^int_s^K-1]  (apologies for pseudo-latex, we were unable to successfully parse this equation with openreview's markdown&latex versions).
>
>
> Regarding the output of the function distribution $\mathcal{D}$: we agree with RRxN6 that sampling a task-encoding parameter vector $\tau \in \mathcal{T}$ corresponds to a distribution of tasks/of POMDPs. However, to simplify notations, and for compactness, we decided to refer to such a distribution as simply a stochastic task. In the updated manuscript, the new “Continuous Task Space” paragraph clearly introduces that $\tau$ is a distribution for which we decide to refer as a single stochastic task.

---

> ### Author Response · Authors · 2022-10-07
> **Answer to Reviewer RxN6 (2/2)**
>
> * * * * *
>
> *R: “The instantiated method depend on a certain number of design choices and hyperparameters. I guess the authors have tried different variations of their method. I would have liked them to be better justified and validated. For instance, how did you choose the values for the threshold \delta_{LP}, for \rho_{high} when combining ALP-GMM with IN, or for the pre-training budget? What values were tried? How sensitive are the results to those results? Since running the experiments is costly, it would not be practical to do that for a new a domain if the results are highly dependent on the choices of those hyperparameters.”*
>
> Thanks for pointing out this lack of details regarding hyperparameter choices. We updated our manuscript to feature an “AGAIN details and hyperparameter choices” paragraph in the AGAIN appendix. Given that multiple hyperparameters had to be set and that our experimental budget was limited, we relied on selecting them either by relying on previous works with similar setups (e.g. we re-used most of ALP-GMM parameters from its original paper) or through informal searches over the course of our experimental campain.  $\delta_{LP}$ is a normalized LP value that can go from 0 (all Gaussians accepted) to 1 (all Gaussians rejected). We set its value to 0.2 as a way to reject any zero-LP Gaussians (or low-LP Gaussians), even if LP estimates for such Gaussians are noisy and go slightly over 0. We tried several values between 0.05 and 0.3 (on former AGAIN versions) and did not observe significant differences. For $\rho_{high}$, i.e. the percentage of random task sampling when using ALP-GMM for pre-training, we initially used 20% as in the ALP-GMM paper. We then tried lowering it to 10%, and observed slightly better performances without exploration issues, and therefore used it for all of our experiments. We did not try other values. Regarding the pre-training budget, we believe it is indeed problem-dependent (we discuss this limitation in the discussion section) since students need to pre-train long-enough to acquire separable competence profiles. In our case we set it to 20% of the training budget in our Walker-Climber experiments (2M episodes) based on both reported performance curves from the ALP-GMM paper in related Box2D setups and from initial experiments with classical ACL teachers in the Walker-Climber environment. We also tried 10% of the training budget, but decided not to use this value as it led to less separable competence profiles. As a whole, we did not experience many difficulties while setting up these hyperparameters, which is encouraging regarding the transfer potential to new domains. However we do agree that cross-domain experiments would have been beneficial to manuscript. We updated the discussion section to mention this future direction.
>
> * * * * *
>
> *R: “In the single student case, I believe that the authors didn't explain how they extract curriculum priors from the training history of the first run. To fully understand and assess this part, I think this information should be provided.”*
>
> Thanks for mentioning this lack of information. We updated this experimental section to feature the information. We extract curriculum priors using the same procedure as in AGAIN (without the student selection part), i.e. by keeping a curated list of Gaussians with $LP_{ti} > \delta_{LP}$, using the same $\delta_{LP}=0.2$ value as in our Classroom Teaching experiments in the Walker-Climber testbed.
>
> * * * * *
>
> *R:”Experimental evaluation is a bit limited”*
>
> We believe to have proposed a thorough evaluation of our Meta-ACL approach (as noted by RQFDk), featuring extensive analysis on synthetic task spaces along with multiple experiments on locomotion environments, which is a form of testbed extensively used in the litterature, e.g. [1-4]. We compared our approach to multiple baselines such as expert made curriculum (Oracle), recent ACL methods (ALP-GMM, ADR), along with multiple AGAIN variants (AGAIN_GT, AGAIN_RND, and IN variants featured in the appendix).
>
> That being said, we do agree with RRxN6 that additional experiments on other testbeds would be beneficial in future work (we updated the discussion to mention this potential future work).
>
> [1] Carlos Florensa et al. Automatic goal generation for reinforcement learning agents. ICML, 2018
>
> [2] Rui Wang, Joel Lehman, Jeff Clune, and Kenneth O. Stanley. Paired open-ended trailblazer (POET):
> endlessly generating increasingly complex and diverse learning environments and their solutions. arXiv, 2019.
>
> [3] Rémy Portelas, Cédric Colas, Katja Hofmann, and Pierre-Yves Oudeyer. Teacher algorithms for curriculum learning of deep rl in continuously parameterized environments. CoRL, 2019.
>
> [4] Rui Wang, Joel Lehman, Aditya Rawal, Jiale Zhi, Yulun Li, Jeffrey Clune, and Kenneth O. Stanley. Enhanced
> POET: open-ended reinforcement learning through unbounded invention of learning challenges and their
> solutions. ICML 2020

---

### Review · Reviewer_ecKL · 2022-10-03

**Summary Of Contributions:**

The paper aims at leveraging procedural task generation systems for Automatic Curriculum Learning (ACL) to adapt the task sampling in order to improve learning. The particularity of the paper is to consider what the authors call "Meta-ACL", which formalize the curriculum generation to an (unknown) distribution of learners. They show the application of their algorithm for some "parkour environments" with learners of varying morphologies.


**Requested Changes:**

The approach does not seem well motivated (e.g. backed up for instance with clear explanations, intuitions or theory). The main problems start according to me from Section 4.1 onwards. As an example, it is written in Section 4.1 the following "The Gaussian from which to sample a new task is chosen proportionally to its mean LP dimension. Task exploration happens initially through a bootstrapping period of random task sampling and during training through residual random task sampling. See appendix A for a detailed description with pseudo-code.". From that sentence it is unclear what the motivation for doing so is. In addition, it is also unclear what the "mean LP dimension" and "residual random task sampling" are (it seems that many important details are given in the appendix to understand what's actually done).

On the formal side, there are different elements in the paper that are not clear. Here is a list of a few of them:
- The formalization of a task space could be given more formally. This might be important especially because the paper mentions "continuous task space". We can understand from the paper that it refers to "task generation systems controlled through complex continuous parameter spaces" as mentioned in the abstract, but it might be good to formalize this more clearly by writing that the distribution of the transition function and reward function depends on some underlying latent variables (which can and should be precisely defined mathematically).
- Some elements are not always clearly defined. For instance when adding subscripts, super scripts, it isn't always clear what the new element is (e.g. Equation 2). Another example is between Equation 2 and 3 in the sentence "f is a function extracting curriculum priors X over a history $\mathcal H^{\mathcal S}$ of past $K$ student trainings, resulting from the scaffolding of K previous students with an ACL or Meta-ACL policy". What is exactly "curriculum priors": e.g. what is the space (vector in $\mathbb R^n$, ...)? The term scaffolding is also not very clear to get a better idea on exactly what $X$ is.
- I could not understand what "LP" stands for until I realized that "learning progress" that is also used in a few places in the paper is likely a good candidate for what the authors meant. I don't think LP is introduced formally anywhere in the paper.



**Strengths And Weaknesses:**

Strengths:
- The paper tackles a novel area of research that can be of interest.
- The paper is globally well-written, at least for the first few sections.

Weaknesses:
- The technical parts are not clear.
- While the overall motivation for the setting is quite clear from the introduction, the formalization of the setting and even more of the approach should be improved. The elements that should be improved are provided below.

---

> ### Author Response · Authors · 2022-10-07
> **Answer to Reviewer EcKL (1/2)**
>
> We thank the reviewer for their valuable comments and suggestions. We were pleased to read that they acknowledge the novelty and relevance of our work.
>
> Similarly to other reviewers, RecKL raised concerns about the writing of our manuscript. Based on all reviews, we submitted a revision of our initial submission, in which each modification/addition in terms of text has been highlighted in red to simplify the review process. In the following paragraphs, we address each of RecKL comments with modifications of our initial submission.
>
> * * * * *
>
> *R: “The approach does not seem well motivated (e.g. backed up for instance with clear explanations, intuitions or theory). The main problems start according to me from Section 4.1 onwards. As an example, it is written in Section 4.1 the following "[...]”. From that sentence it is unclear what the motivation for doing so is. In addition, it is also unclear what the "mean LP dimension" and "residual random task sampling" are (it seems that many important details are given in the appendix to understand what's actually done).”*
>
> *R: “I could not understand what "LP" stands for until I realized that "learning progress" that is also used in a few places in the paper is likely a good candidate for what the authors meant. I don't think LP is introduced formally anywhere in the paper.”*
>
> Thanks for pointing out this lack of explanations in section 4, and the need to move appendix material into the main body of the paper. Similar concerns were raised by RQFDk and RRxN6. As such, we decided to perform a major update on this section (available in the updated manuscript). In summary, we performed the following modifications:
>
> * We significantly increased section 4.1 on ALP-GMM, by both adding and updating content from the appendix. The main body now contains the pseudo-code of ALP-GMM and 2 visualizations to help understand the algorithm. We added two paragraphs to detail the two main components of ALP-GMM: “Absolute LP computation” and “GMM fitting and sampling”. The section now features an equation describing Gaussians (and their local LP measure), along with details on our sampling procedure
>
> * We added details on section 4.2 to improve the description of how students are selected after the pre-training phase, and significantly extended section 4.4 with important AGAIN descriptions initially left in the appendix. We also moved AGAIN pseudo code into the main body of the paper.
>
> Sincere apologies regarding the missing LP acronym (we now introduce it early in the introduction section). Regarding the lack of definition about what is the “mean LP dimension”, we believe our updated section 4.1 now better present this term. Beyond its general meaning, we use the term “learning progress” for two specific components of our algorithms:
>
> (1) The task-specific Absolute LP estimates $ALP_{\tau}$ computed by ALP-GMM, used to create a database of task-ALP pairs, and
>
> (2) The local (Gaussian-specific) Absolute LP →Given a database of task-ALP pairs, ALP-GMM's main mechanism is to fit a GMM on the concatenated space of tasks and ALP, i.e. a dataset of vectors of $n+1$ dimensions (given a $n$-dimensional task space). Using such a fitting process -- relying on task-specific $ALP_{\tau}$ estimates -- one can then obtain a set of $k$ Gaussians $\Bigl(\mathcal{N}(\mu_{i},\Sigma_{i})\Bigl)_{i=1}^k$
>
> with $\mu_{i} \in \mathbb{R}^{n+1}$. Given this formulation, one can interpret the mean ALP dimension of $\mu_{i}$ as a local aggregated ALP measure, which we propose to refer to as $ALP_i$. It is this $ALP_i$, $LP_i$ for short, that appears inside the $X_ {raw}$ equation of section 4.3 (the sub-script $t$ indicates the position of the GMM in the list). This $LP_i$ value of a Gaussian $i$ constitutes a noisy measure of the (absolute) learning progress expected if selecting a task by sampling in $i$. This is what we call the “mean LP dimension”
>
> **About residual sampling** Finally, by “residual random task sampling”, we just mean that in addition to its main GMM-based task sampling, ALP-GMM occasionally samples random tasks to detect potential new high-ALP subspaces (10% of random task sampling in our experiments with ALP-GMM). The updated section 4.1 now better introduces this term (it also features a clear explanation of ALP-GMM’s task sampling mechanism in the “GMM fitting and sampling” paragraph).

---

> ### Author Response · Authors · 2022-10-07
> **Answer to Reviewer EcKL (2/2)**
>
> * * * * *
> *R: “The formalization of a task space could be given more formally. This might be important especially because the paper mentions "continuous task space". We can understand from the paper that it refers to "task generation systems controlled through complex continuous parameter spaces" as mentioned in the abstract, but it might be good to formalize this more clearly by writing that the distribution of the transition function and reward function depends on some underlying latent variables (which can and should be precisely defined mathematically).”*
>
> Thanks for spotting this lack of details. We updated the formalization to feature a “Continuous Task Space” paragraph. We use the term continuous task space to refer to a parameter space $\mathcal{T} \subset \mathbb{R}^n$ encoding the transition and reward functions along with the initial state distribution of Partially Observable Markov Decision Processes: any $\tau \in \mathcal{T}$ encodes the procedural generation of a (stochastic) environment, a.k.a. task. As in the ALP-GMM paper (from which we extend the teacher-student framework), minimal assumptions are made over $\mathcal{T}$: the difficulty profile of $\mathcal{T}$ is unknown and therefore assumed to be a piece-wise smooth function, potentially containing large unfeasible or trivial subspaces (w.r.t. to considered learners).
>
> * * * * *
>
> *R: “Some elements are not always clearly defined. For instance when adding subscripts, super scripts, it isn't always clear what the new element is (e.g. Equation 2). Another example is between Equation 2 and 3 in the sentence "f is a function extracting curriculum priors X over a history HS of past K student trainings, resulting from the scaffolding of K previous students with an ACL or Meta-ACL policy". What is exactly "curriculum priors": e.g. what is the space (vector in Rn, ...)? The term scaffolding is also not very clear to get a better idea on exactly what X is.”*
>
> Thanks for pointing out this lack of clarity, related to RxN6 comments on our formalization. We decided to rework our formalization section to drop unnecessary complexities and more accurately introduce components.
>
> Regarding your concerns, we now believe calling $f$ a function is not accurate: it is rather an algorithm. We switched the term “scaffolding” for “teaching”, which is more accurate and less ambiguous. We also added examples of curriculum priors (to help conceptualize the term).
>
> This leads to the following update of the sentence: “$f$ is an algorithm extracting curriculum priors $X$ over a history $\mathcal{H}^{\mathcal{S}}$ of past K student trainings, resulting from the teaching of $K$ previous students with an ACL or Meta-ACL policy. We denote by curriculum priors any form of information relevant to augment a curriculum generation mechanism, e.g. a set of important tasks to focus on (i.e. a list of vectors) or -- as in our experiments -- a list of Gaussians.”

---

### Review · Reviewer_QFDk · 2022-10-04

**Summary Of Contributions:**

This paper formalizes Meta-Automatic Curriculum Learning, which is the idea of generating a custom curriculum for each learning agent in a population of agents (summarized in Equation 2 and Figure 1). It then proceeds to introduce a new algorithm, AGAIN. In this algorithm, given a history of previously trained students, and a new student, AGAIN first trains this new student with a different algorithm, ALP-GMM; it then presents a set of tasks to the new student and uses the student’s performance to generate a competence profile. This competence profile is used as a model of the agent, and AGAIN uses k-nn to map the new student to a former student (from the given history of previously trained students) with similar competence profile. AGAIN then generates a curriculum similar to the matched students’, further training the new student.

The paper compares the performance of the proposed algorithm with several variations of it, ablating the different components. Experiments are performed in a simple toy environment and variants of a Walker-Climber Environment.


**Broader Impact Concerns:**

I have no concerns.


**Requested Changes:**

As I wrote above, the paper introduces a method for a well-defined problem and it performs sensible experiments to support its claims. In terms of evaluation criteria, it seems to me that there is proper evidence to support the claims made in the submission, from what I understood or inferred, and that it is a topic some individuals in TMLR’s audience would be interested in knowing. Nevertheless, I do think the paper has severe presentation issues, and it needs to be addressed. The paper would be drastically improved if the text was adjusted to answer the questions I enumerated above in bullet points. I acknowledge the paper will be longer, but it will be substantially better and easier to follow.

**Strengths And Weaknesses:**

This paper does a good job discussing the problem they are tackling and its experiments seem sound, with a proper number of independent trials to allow for significant claims about performance. The results presented seem convincing to me, although the topic in the paper is not in my research area and I don’t feel able to assess whether there are any important baselines missing.

To me, the main weakness of the paper is presentation. Simply put, the paper is hard to read. The paper uses too many acronyms throughout, sometimes shortening words that do not even need to be shortened to make the sentence fit in a specific number of lines. The paper glosses over important algorithmic details (sometimes they are in the appendix, sometimes they are not), and uses expressions that sometimes I don’t understand. I also don’t see how evaluating the proposed algorithm is a contribution (the third claimed contribution). The contribution is the algorithm, and the only reason it is the algorithm is because it is demonstrated that it is either theoretically or empirically relevant.

Below I the questions I had as I was reading the paper:
* What’s a “non-resetable student”, or a “continuous task space” (Section 3)? I don’t think these terms were defined.
* Right after equation 1, it says E is the “episode budget”. What does that mean? The number of steps within an episode? The number of episodes the agent is allowed to interact with the environment? From the equation it seems it is the latter, but the wording is not clear.
* What is LP in Section 4? Eventually I understood it is learning progress, but to the best of my knowledge it is only defined in Appendix A. Hopefully by now one can appreciate how hard it is to read a paper like this when acronyms are used so often and sometimes are not defined.
* Still talking about Appendix A, theoretically Appendix A explains ALP-GMM, which is absolutely central to this paper. I couldn’t understand ALP-GMM from Appendix A, even though there is even a pseudo-code in there. It still feels too high level. What are the equations for the GMM? What is the Akaike’s Information Criterion? Obviously, this is background material, not the paper’s contribution, but because the paper builds so heavily from it, I think it should be presented in the main paper and in more detail.
* I don’t understand what “episodic reward” means in Appendix A. Is it the return, and the reason it is called reward is because it is a bandits problem? The same happens in the main text.
* AGAIN is the main contribution of the paper, but it is actually only presented in the Appendix. I don’t understand why it has to be presented so concisely in the paper. Importantly, the same criticism above applies here, without the equations, it is not clear to me whether someone can actually replicate this algorithm. It seems all we have to understand the algorithm are several algorithmic boxes in the appendix in a high-level.
* It is not clear to me what is the role of Step 3 in Section 4.2.
* What is the “Learning Progress of a Gaussian”, as described in Section 4.3. How is that an utility function? The paper uses the term
 $LP_{ti}$ value, but I couldn’t find where this value was formally defined.
* Shortly, the paragraph before Section 5 is the one that introduces AGAIN. It seems to me it should be drastically expanded, to actually present the algorithm in the main paper.
* I don’t understand the toy domain introduced. Is the task the same thing as choosing a start state? Is this a bandits problem? Is there learning going on? What’s the action set in this problem? Is it an MDP? I couldn’t figure this out from the paper.
* It is said that the “length of the initial pre-training period” “is crucial and must be carefully selected by the experimenter”. Do we have this data available anywhere? It would be interesting to understand the impact of this parameter in the performance.

---

> ### Author Response · Authors · 2022-10-07
> **Answer to Reviewer QFDk (1/3)**
>
> We thank the reviewer for their valuable comments and suggestions. We were pleased to see that they acknowledge the significance of our experimental analysis.
>
> We acknowledge RQFDk analysis, which is similarly rooted as those of reviewers ecKL and RxN6: the current version of the manuscript could be drastically improved in terms of writing. Based on all reviews, we submitted a revision of our initial submission, in which each modification/addition in terms of text has been highlighted in red to simplify the review process. In the remainder of this answer, we address each of the points that were raised by RQFDk.
>
> * * * * *
>
> *R: “What’s a “non-resetable student”, or a “continuous task space” (Section 3)? I don’t think these terms were defined.”*
>
> These terms are indeed poorly introduced in the current version, thanks for spotting it. About “non-resetable student”: our intention was to mention that it is not possible to re-initialize or restore a student's knowledge state in the middle of training (which makes ACL and Meta-ACL research challenging). For instance, we do not assume it is possible to use backed-up neural network parameters in the advent of catastrophic forgetting. From a teacher/ACL perspective, it means it is not possible to teach multiple times the same student, i.e. the optimization of the task sampling policy must be done in a single rollout (unlike classical behavioral policy learning in Deep RL, which predominantly assumes resetable environments).
>
> Regarding the term “continuous task space”, we use it to contrast our setup with works studying ACL on a finite set of discrete tasks: in our work, we consider a set of task’s parameters $\mathcal{T} \subset \mathbb{R}^n$ encoding an infinity of task. In the updated manuscript we slightly rephrased the abstract to clearly introduce the term “continuous task spaces” early on, and we added a Continuous Task Space paragraph inside our formalization section to describe it formally. We also updated the formalization section to define the term “non-resetable student”.
>
> * * * * *
>
> *R: “Right after equation 1, it says E is the “episode budget”. What does that mean? The number of steps within an episode? The number of episodes the agent is allowed to interact with the environment? From the equation it seems it is the latter, but the wording is not clear.”*
>
> Correct, it is the total number of episodes the agent is allowed to interact with the environment. We updated the manuscript to make this clear.
>
> * * * * *
>
> *R: “Appendix A explains ALP-GMM, which is absolutely central to this paper. I couldn’t understand ALP-GMM from Appendix A, even though there is even a pseudo-code in there. It still feels too high level. What are the equations for the GMM? What is the Akaike’s Information Criterion? Obviously, this is background material, not the paper’s contribution, but because the paper builds so heavily from it, I think it should be presented in the main paper and in more detail.”*
>
> *R: “AGAIN is the main contribution of the paper, but it is actually only presented in the Appendix. I don’t understand why it has to be presented so concisely in the paper. Importantly, the same criticism above applies here, without the equations, it is not clear to me whether someone can actually replicate this algorithm. It seems all we have to understand the algorithm are several algorithmic boxes in the appendix in a high-level.”*
>
> *R:”Shortly, the paragraph before Section 5 is the one that introduces AGAIN. It seems to me it should be drastically expanded, to actually present the algorithm in the main paper.”*
>
> Thanks for pointing out this lack of details in section 4. Given similar concerns were raised by all reviewers, we decided to perform a major update on this section (available in the updated manuscript). In summary, we performed the following modifications:
>
> * We significantly increased section 4.1 on ALP-GMM, by both adding and updating content from the appendix. The main body now contains the pseudo-code of ALP-GMM and 2 visualizations to help understand the algorithm. We added two paragraphs to detail the two main components of ALP-GMM: “Absolute LP computation” and “GMM fitting and sampling”. The section now features an equation describing Gaussians (and their local LP measure), along with details on our sampling procedure. We also briefly explain what Akaike’s Information Criterion is, i.e. a statistical measure of how accurate the GMM model is w.r.t. to the considered task-ALP pairs to be fitted.
>
> * Following RQFDk suggestions, we also moved AGAIN pseudo code into the main body of the paper, along with important descriptions initially left in the appendix, which we updated and merged to our initial paragraphs (section 4.4). We mainly focused on adding information on how the IN curriculum is built and combined to ALP-GMM in AGAIN. We also improved the description of how students are selected after the pre-training phase (section 4.2).

---

> ### Author Response · Authors · 2022-10-07
> **Answer to Reviewer QFDk (2/3)**
>
> * * * * *
>
> *R: “What is LP in Section 4? Eventually I understood it is learning progress, but to the best of my knowledge it is only defined in Appendix A. Hopefully by now one can appreciate how hard it is to read a paper like this when acronyms are used so often and sometimes are not defined.”*
>
> *R: “What is the “Learning Progress of a Gaussian”, as described in Section 4.3. How is that an utility function? The paper uses the term LPti value, but I couldn’t find where this value was formally defined.*
>
> Sincere apologies for this missing acronym definition and lack of details about technical instantiations of the term learning progress. We updated the manuscript to define this acronym early in the introduction section. The reworked section 4.1 now features enough material to properly understand the notion of “learning progress”. More precisely, beyond its general meaning of “competence improvement”, we use this term for two specific components of our algorithms:
>
> (1) The task-specific Absolute LP estimates $ALP_{\tau}$ computed by ALP-GMM, used to create a database of task-ALP pairs, and
>
> (2) The local (Gaussian-specific) Absolute LP → Given a database of task-ALP pairs, ALP-GMM's main mechanism is to fit a GMM on the concatenated space of tasks and ALP, i.e. a dataset of vectors of $n+1$ dimensions (given a $n$-dimensional task space). Using such a fitting process -- relying on task-specific $ALP_{\tau}$ estimates -- one can then obtain a set of $k$ Gaussians $\Bigl(\mathcal{N}(\mu_{i},\Sigma_{i})\Bigl)_{i=1}^k$
>
> with $\mu_{i} \in \mathbb{R}^{n+1}$. Given this formulation, one can interpret the mean ALP dimension of $\mu_{i}$ as a local aggregated ALP measure, which we propose to refer to as $ALP_i$. It is this $ALP_i$, $LP_i$ for short, that appears inside the $X_ {raw}$ equation of section 4.3 (the sub-script $t$ indicates the position of the GMM in the list). This $LP_i$ value of a Gaussian $i$ constitutes a noisy measure of the (absolute) learning progress expected if selecting a task by sampling in $i$. From this perspective, it can be seen as the (estimated) utility of the Gaussian.
>
> * * * * *
>
> *R: “I don’t understand what “episodic reward” means in Appendix A. Is it the return, and the reason it is called reward is because it is a bandits problem? The same happens in the main text.”*
>
> Yes indeed, we use the term “episodic reward” as a synonym to refer to the return of an episode. We updated the manuscript to better present this term and clearly state it is equivalent to a return.
>
> * * * * *
>
> *R: “It is not clear to me what is the role of Step 3 in Section 4.2.”*
>
> Step 1 & 2 describe our method to select a set of k previously trained students whose initial competence profiles (i.e. the $CP^{pre}_{s^K}$ vector) are similar to the initial competence profile of the currently trained student. In step 3, we then describe how AGAIN selects one student out of these k to extract curriculum data. This selection is based on comparing the post-training performance (i.e. the post-training score $j_s$) of these k students. The student with highest post-training score is selected, as it is the one which improved most across its training and is therefore more likely to hold interesting curriculum data. We updated section 4.2 to better describe this step.
>
> * * * * *
>
> *R:”I don’t understand the toy domain introduced. Is the task the same thing as choosing a start state? Is this a bandits problem? Is there learning going on? What’s the action set in this problem? Is it an MDP? I couldn’t figure this out from the paper.”*
>
> One can see our toy domain as a tool implementing a fake task space along with a simulation of competence improvement from fake policy learners over this space. There is no learning happening, apart from the teacher learning a task sampling policy. For instance, a cell in the task space (a subspace) can be seen as a group of task-encoding parameters creating bipedal locomotion challenges of similar difficulty profiles. Another example would be that these task-encoding parameters correspond to the initial position of blocks in a robotic manipulation environment where blocks must be arranged into a fixed position (in this case choosing a task is indeed like choosing the initial state of an MDP). In practice however, there are no such tasks in our toy domain: the teacher samples a task and immediately observes a (fake) episodic reward based on our simulated student-competence improvement mechanism. We updated our manuscript to improve the clarity of the toy domain’s description.

---

> ### Author Response · Authors · 2022-10-07
> **Answer to Reviewer QFDk (3/3)**
>
> * * * * *
>
> *R:”It is said that the “length of the initial pre-training period” “is crucial and must be carefully selected by the experimenter”. Do we have this data available anywhere? It would be interesting to understand the impact of this parameter in the performance.”*
>
> Regarding the pre-training budget, we believe it is indeed problem-dependent since students need to pre-train long-enough to acquire separable competence profiles. In our case we set it to 20% of the training budget in our Walker-Climber experiments (2M episodes) based on both reported performance curves from the ALP-GMM paper in related Box2D setups and from initial experiments with classical ACL teachers in the Walker-Climber environment. We also tried 10% of the training budget, but decided not to use this value as it led to less separable competence profiles. We added an “AGAIN details and hyperparameter choices” paragraph in the appendix which includes this analysis. As a whole, we did not experience many difficulties while setting up these hyperparameters, which is encouraging regarding the transfer potential to new domains. However we do agree that cross-domain experiments would have been beneficial to the manuscript. We updated the discussion section to mention this future direction.
>
> * * * * *
>
> *R:”I also don’t see how evaluating the proposed algorithm is a contribution (the third claimed contribution). The contribution is the algorithm, and the only reason it is the algorithm is because it is demonstrated that it is either theoretically or empirically relevant.”*
>
> We agree that introducing and analyzing AGAIN are better seen as a single contribution: we updated the manuscript accordingly.

---

### Decision · Action_Editors · 2022-12-12

**Recommendation:** Reject

**Comment:**

Given that the claims don't seem supported well enough by experiments (generality of the method is questionable) and that the audience is restricted by the quality of the writing and clarity of the paper (good knowledge of the curriculum learning literature is required), all the reviewers suggested to reject this paper and encourage the authors to improve the clarity of the paper.

**Audience:**

The reviewers unanimously appreciate that this work could be of interest to the curriculum learning community. Yet, the quality of the writing makes it hard to understand by researchers who are not familiar with that field. Some terms are not defined, definitions are sometimes ambiguous and a lot of questions remain unresponded or the answer is still ambiguous. For this reason, the audience seems a bit too narrow to deserve publication of the paper in its current presentation.

**Claims And Evidence:**

All the reviewers acknowledged that the topic addressed in this paper is interesting, it brings a new perspective on automatic curriculum learning and provides a simple heuristic to solve it. Yet, after the discussion which provided details about the experimental setting, reviewers also acknowledged that they are not convinced that the evidence that support the claim are general enough and would not stand in more complex situations as claimed by the authors. Especially, given that learning only happens on one side of the system (no learning in the students), it is unclear that the experiments are providing enough insights about how well the method would behave in other settings. In that respect, the paper doesn't match the TMLR requirements for publications.

---

> ### Author Response · Authors · 2022-12-16
> **Comment regarding paper decision**
>
> While we respectfully acknowledge the decision of the review committee, we would like to discuss some of the arguments that were raised for the rejection.
>
> > Action Editors: *"after the discussion which provided details about the experimental setting, reviewers also acknowledged that they are not convinced that the evidence that support the claim are general enough and would not stand in more complex situations as claimed by the authors. Especially, given that learning only happens on one side of the system (no learning in the students), it is unclear that the experiments are providing enough insights about how well the method would behave in other settings."*
>
> We respectfully disagree with the statement that in our experiments "*learning only happens on one side of the system*" . While this is true for our preliminary experiments on a toy environment in section 5.1 (the student is simulated), all experiments presented in section 5.2 and 5.3 do feature learning both on the teacher/ACL side and on the student side: teachers sample tasks for PPO agents which are learning a locomotion policy from scratch over the course of the curriculum generation. Section 5.2 and 5.3 precisely showcases that the performance claims and trends of our proposed Meta-ACL  methods do scale to more complex situations, i.e. complex locomotion learning with DRL agents on non-trivial parametric task spaces.
>
> > Action Editors: *"the quality of the writing makes it hard to understand by researchers who are not familiar with that field. Some terms are not defined, definitions are sometimes ambiguous and a lot of questions remain unresponded or the answer is still ambiguous"*
>
> We do agree that our initial submission was lacking important definitions and that the overall writing clarity regarding our framework and especially methods section could be improved. In particular both RQFDk and RecKL rightfully complained about crucial material being left to the appendix. Based on these helpful remarks and suggestions for improvements, we submitted a significantly updated version of our manuscript, which features 3 pages of additional material in the main body of the paper (consisting both of reworks of our initial appendix material and of new methodological details based on the review discussion).
>
> We are sorry to learn that reviewers and/or the action editor(s) judged that "*a lot of questions remain unresponded or the answer is still ambiguous*". We would have been very happy to keep the discussion going to elucidate these issues. Unfortunately we did not receive any reaction to our initial reply and updated manuscript, both sent to reviewers the 7th of October. We contacted the action editor one month later to ask whether we should send a gentle reminder to reviewers (if there was a hard deadline for the end of discussion), but did not receive any reply as well.
>
> As of today, we are still wondering whether there might have been some issue in the comment publishing system of open-review. Could the action editor(s) acknowledge whether our replies and updated manuscript were accessible and taken into account by all reviewers ?
>
> To conclude, in this comment, we answered the reviewers/editor(s) concern about the lack of learning in our experiments: we do feature learning students in section 5.2 and 5.3. Given that our initial replies to reviewers and updated manuscript have yet to be discussed, would the action editor(s) be willing to let us re-submit our work ?